# Systematic studies of all PIH proteins in zebrafish reveal their distinct roles in axonemal dynein assembly

Hiroshi Yamaguchi[1,2], Toshiyuki Oda[3], Masahide Kikkawa[1]*, Hiroyuki Takeda[2]*

[1]Department of Cell Biology and Anatomy, Graduate School of Medicine, The University of Tokyo, Tokyo, Japan; [2]Department of Biological Sciences, Graduate School of Science, The University of Tokyo, Tokyo, Japan; [3]Department of Anatomy and Structural Biology, Graduate School of Medicine, University of Yamanashi, Yamanashi, Japan

**Abstract** Construction of motile cilia/flagella requires cytoplasmic preassembly of axonemal dyneins before transport into cilia. Axonemal dyneins have various subtypes, but the roles of each dynein subtype and their assembly processes remain elusive in vertebrates. The PIH protein family, consisting of four members, has been implicated in the assembly of different dynein subtypes, although evidence for this idea is sparse. Here, we established zebrafish mutants of all four PIH-protein genes: *pih1d1*, *pih1d2*, *ktu*, and *twister*, and analyzed the structures of axonemal dyneins in mutant spermatozoa by cryo-electron tomography. Mutations caused the loss of specific dynein subtypes, which was correlated with abnormal sperm motility. We also found organ-specific compositions of dynein subtypes, which could explain the severe motility defects of mutant Kupffer's vesicle cilia. Our data demonstrate that all vertebrate PIH proteins are differently required for cilia/flagella motions and the assembly of axonemal dyneins, assigning specific dynein subtypes to each PIH protein.
DOI: https://doi.org/10.7554/eLife.36979.001

*For correspondence:
mkikkawa@m.u-tokyo.ac.jp (MK);
htakeda@bs.s.u-tokyo.ac.jp (HT)

Competing interests: The authors declare that no competing interests exist.

## Introduction

Motile cilia/flagella are hair-like organelles that project from various types of eukaryotic cells. In humans, malfunctions of motile cilia often cause primary ciliary dyskinesia (PCD), a syndrome characterized by recurrent respiratory infections, male infertility, hydrocephalus, and inversion of visceral laterality (*Knowles et al., 2013*; *Brown and Witman, 2014*). Motile cilia have a microtubule-based structure called an axoneme, consisting of nine peripheral doublet microtubules (DMTs) with or without central-pair microtubules (so called 9 + 2, 9 + 0, respectively). Ciliary motility is driven by axonemal dyneins, which have multiple subtypes such as outer arm dyneins (OADs) and seven different types of inner arm dyneins (IADs; IAD a to g; *Kagami and Kamiya, 1992*). Biochemical analyses of green algae *Chlamydomonas* revealed that each axonemal dynein consists of multiple subunits (*Hom et al., 2011*; *Sakato and King, 2004*); OAD is composed of three heavy chains (α-, β-, and γ-HC), two intermediate chains (IC1 and IC2), and ten light chains. Six types of IADs (IAD a, b, c, d, e, and g) have single HCs with several light chains such as p28, centrin, and actin. IAD f has two heavy chains (f α- and f β-HC), four intermediate chains, and five light chains.

In the process of ciliary construction, axonemal dyneins and all other large ciliary molecules are synthesized in the cytoplasm and undergo gated entry into the ciliary compartment (*Takao and Verhey, 2016*). In the cytoplasm, the components of OADs and IADs are detected as preassembled complexes, rather than individual components (*Fok et al., 1994*; *Fowkes and Mitchell, 1998*; *Viswanadha et al., 2014*). This cytoplasmic preassembly of axonemal dyneins requires various

**eLife digest** Many cells have long, thin structures called cilia on their surface, some types of which can beat back and forth. This beating motion has many roles; for example, cilia on the cells that line the lungs help to sweep out debris, and the tails of sperm beat to move them forward.

A structure called the axonemal dynein complex at the core of the cilia generates the beating motion. When the cell makes new cilia, it assembles the complexes in the main body of the cell and then transports them to the right place, like erecting a prefabricated building. Various proteins help to assemble the complexes, of which there are more than eight types. However, the identities of all of these proteins, and their roles in constructing specific axonemal dynein complexes, is not fully known.

Studies in algae have suggested that a family of proteins known as PIH (short for protein interacting with Hsp90) helps to construct axonemal dynein complexes. Zebrafish – which share many of the same protein-encoding genes as humans – produce four PIH family proteins. To investigate the roles that each of these proteins play, Yamaguchi et al. used genetic engineering to create four zebrafish mutants that were each unable to produce a different PIH protein.

A technique called cryo-electron microscopy enabled the axonemal dynein complexes in the tails of the sperm produced by the zebrafish to be visualized. The sperm from each mutant lacked specific axonemal dynein complexes, revealing that each PIH protein assembles different complexes. The sperm also had difficulties moving. Yamaguchi et al. examined this movement to deduce how specific complexes affect the ability of the sperm to beat their tails.

Further work on how PIH proteins interact with the axonemal dynein complexes will help us to understand how cells make cilia, and what happens when this process goes wrong. This could ultimately help us to treat genetic disorders known as ciliopathies, which arise when cilia do not develop normally.

DOI: https://doi.org/10.7554/eLife.36979.002

proteins collectively called dynein axonemal assembly factors (DNAAFs; *Kobayashi and Takeda, 2012*; *Mitchison et al., 2012*). As cilia/flagella require multiple types of axonemal dyneins for their motions (*Kamiya, 1995*), proper assembly of each dynein complex is essential for ciliary motility. However, the assembly processes of each dynein subtype and their roles in cilia/flagella motions remain elusive in vertebrates.

The PIH protein family has been implicated in the preassembly of different subsets of axonemal dyneins. The PIH protein, which contains a PIH1-domain, was first identified in budding yeast (*Saccharomyces cerevisiae*) as an interactor of HSP90 and named as Pih1 (Protein Interacting with HSP90; also known as Nop17; *Zhao et al., 2005*; *Gonzales et al., 2005*). Yeast Pih1 is required for the assembly of various multi-subunit protein complexes but is not involved in the assembly of axonemal dyneins, as yeast do not have a cilium. In vertebrates, there are four PIH proteins: PIH1D1, PIH1D2, KTU/DNAAF2, and PIH1D3/TWISTER, and PIH1D1 is the orthologue of yeast Pih1. Similar to yeast, the human PIH1D1 is a subunit of the R2TP complex (RUVBL1, RUVBL2, RPAP3/Tah1, and PIH1D1), which interacts with HSP90 to promote assembly of various protein complexes for cellular activities such as box C/D snoRNP and RNA polymerase II (*Kakihara and Houry, 2012*).

KTU/DNAAF2 is the first protein that was identified as a DNAAF (*Omran et al., 2008*). Genetic and biochemical analyses of KTU/DNAAF2 in medaka (Japanese killifish), human, and *Chlamydomonas* revealed that KTU/DNAAF2 is required for the assembly of OAD and a subset of IADs (*Omran et al., 2008*). Subsequently, the function of *Chlamydomonas* MOT48, a possible orthologue of vertebrate PIH1D1, was reported; MOT48 is one of three PIH proteins in *Chlamydomonas* (MOT48, PF13/KTU, and TWI1) and is involved in the assembly of another subset of IADs (*Yamamoto et al., 2010*). These pioneering studies proposed that preassembly of different subsets of axonemal dyneins is mediated by distinct PIH proteins. However, evidence for this hypothesis is sparse, due to a lack of systematic studies of all PIH proteins. Although PIH1D3/TWISTER recently turned out to be one of the DNAAFs (*Dong et al., 2014*; *Paff et al., 2017*; *Olcese et al., 2017*), the function of vertebrate PIH1D1 and PIH1D2 has not been addressed in terms of ciliogenesis and ciliary motility. Furthermore, two *Pih1d3* paralogues in mice, *Pih1d3* and *Twister2*, are differently

expressed in ciliary/flagellar organs (*Pih1d3* for testis, while *Twister2* for both testis and the others; *Dong et al., 2014*), suggesting a divergence of their functions.

In this study, we performed systematic and functional analyses of all four PIH genes (genes encoding PIH proteins) by generating zebrafish (*Danio rerio*) mutants of each PIH gene. We compared the functions of all PIH proteins in one platform, zebrafish, because functional divergence of PIH proteins among organisms is highly probable. Although zebrafish is often used to analyze the functions of PCD related genes, the detailed structure of their cilia/flagella has not been studied so far. We applied cryo-electron tomography (cryo-ET) for the first time to zebrafish sperm, which enabled us to observe the detailed structure of wild-type and mutant axonemal dyneins. Mutations of each PIH gene caused defects of different subtypes of axonemal dyneins, which was correlated with abnormal sperm motility. Interestingly, some mutants showed different phenotypes of ciliary motility between sperm flagella and Kupffer's vesicle cilia. Together with different expression patterns of various DNAH (dynein axonemal heavy chain) genes, we also discuss the organ-specific compositions of axonemal dyneins assembled by PIH proteins. This is the first report that shows all vertebrate PIH proteins as DNAAFs, assigning their functions to specific types of axonemal dyneins and to cilia/flagella motions. Our data provide evidence for the above-mentioned hypothesis that the cytoplasmic assembly of different dynein subtypes is mediated by distinct PIH proteins.

## Results

### Generation of zebrafish mutants of the *pih1d1*, *pih1d2*, *ktu* and *twister* genes

To find all PIH proteins encoded in the zebrafish genome, we performed BLASTp search with the consensus sequence of PIH proteins as a query. Although teleost fish are known to have undergone an additional genome duplication (*Kasahara et al., 2007*), only four hits were obtained, similar to the human genome and consistent with a previous report (*Yamamoto et al., 2010*). We thus conclude that zebrafish has four PIH proteins: Pih1d1, Pih1d2, Ktu, and Twister. Their domain structures are well conserved among vertebrates (*Figure 1A*; *Figure 1—figure supplement 1A,B*).

Transcripts of these four PIH genes were all detected in ciliated organs such as Kupffer's vesicle, floor plate, otic vesicle and pronephric duct (*Figure 1D–G*), suggesting their involvement in ciliary functions. The transcript of *pih1d1* was also detected in the whole body of 12 hpf (hours post-fertilization) embryos (*Figure 1D*; black asterisk), which is consistent with the reported cellular functions of human PIH1D1 (*Kakihara and Houry, 2012*). *ktu* was also expressed in brain rudiments at 32 hpf (*Figure 1D*; black arrowhead). In mouse, KTU function in brain was reported for ciliated ependymal cells (*Matsuo et al., 2013*). These results suggest that *pih1d2*, *ktu*, and *twister* have cilia-specific functions, while *pih1d1* has ubiquitous cellular functions in addition to ciliary function.

Since DNAAFs including PIH proteins are known to be localized to the cytoplasm (*Kobayashi and Takeda, 2012*), we examined the subcellular localizations of zebrafish PIH proteins by immunoblot analysis, using specific polyclonal antibodies made in this study. All PIH proteins were detected in both testis lysate and sperm lysate. However, when spermatozoa were fractionated into sperm heads and flagella, PIH proteins were detected only in the sperm head fraction, indicating that like other DNAAFs, zebrafish PIH proteins are indeed specifically present in the cytoplasm, but not in the flagellar compartment (*Figure 1C*; asterisks).

To analyze the functions of the four PIH proteins in zebrafish, we generated mutant alleles by genome-editing with TALEN (*pih1d1* and *pih1d2*) or CRISPR/Cas9 (*ktu* and *twister*) (*Figure 1B*; *Figure 1—figure supplement 1C,D*). Immunoblot analysis confirmed that all mutations resulted in a null of each PIH protein (*Figure 1—figure supplement 1E*). Since homozygous mutants of each PIH gene were viable, we established homozygous mutant lines. These were used in the following experiments.

### *pih1d1*[-/-], *twister*[-/-], and double mutant of *pih1d2*[-/-];*ktu*[-/-] showed abnormal sperm motility

Ciliary functions of PIH genes were first examined by observing the motility of mutant spermatozoa using a high-speed camera. Spermatozoa whose heads were attached to a coverslip were selected and subjected to the analyses of beating frequencies and waveforms (*Video 1*). In *pih1d1*[-/-],

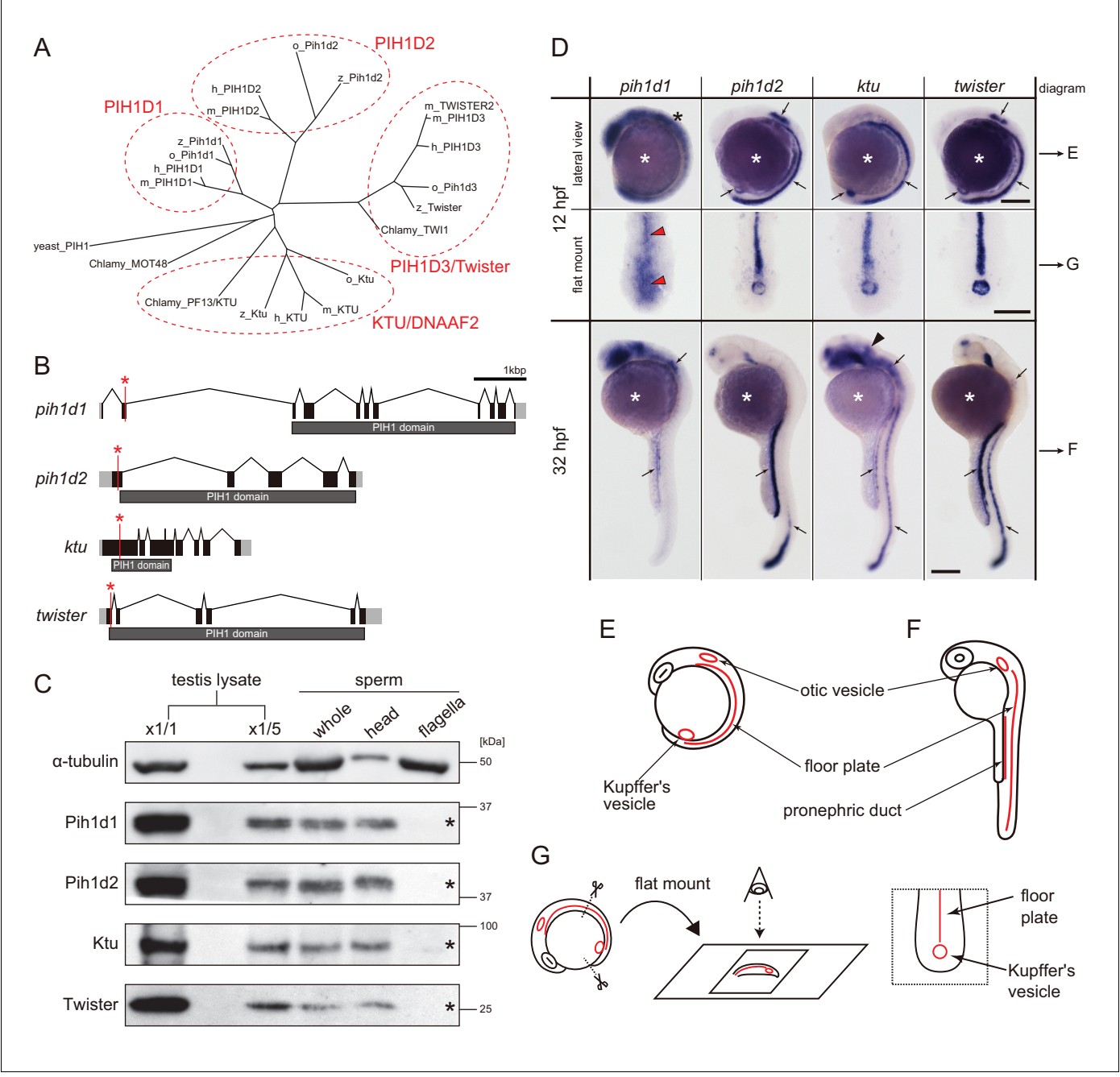

**Figure 1.** Zebrafish PIH genes, *pih1d1*, *pih1d2*, *ktu*, and *twister*, were expressed in ciliated organs. (**A**) Phylogenetic tree of PIH proteins in yeast, *Chlamydomonas*, and vertebrates. Amino acid sequences were aligned by MAFFT program with FFT-NS-i option (**Katoh and Standley, 2013**) and evolutionary distances were calculated using neighbor-joining (**Saitou and Nei, 1987**). yeast, *Saccharomyces cerevisiae*; Chlamy, *Chlamydomonas reinhardtii*; z, zebrafish (*Danio rerio*); o, medaka (*Oryzias latipes*); m, *Mus musculus*; h, *Homo sapiens*. (**B**) Genomic organization of PIH genes in zebrafish. Black: exons. Gray: untranslated regions. Red asterisks indicate the target sites of our genome-editing. (**C**) Immunoblot of PIH proteins. None of the PIH proteins were detected in the sperm flagella fraction (asterisks). α-tubulin: control. (**D**) Whole-mount in situ hybridization of PIH genes. Arrows in lateral views indicate expression of PIH genes in ciliated organs (Kupffer's vesicle, floor plate, otic vesicle, and pronephric duct). Flat mount preparations show dorsal views of the posterior regions of 12 hpf embryos. *pih1d1* was ubiquitously expressed in 12 hpf embryos (black asterisk) containing Kupffer's vesicle and floor plate (red arrowheads). *ktu* was also expressed in brain rudiments at 32 hpf (black arrowhead). White asterisks: non-specific staining of yolk. Scale bars: 200 μm. (**E and F**) Diagrams of zebrafish embryos at (**E**) 12 hpf and (**F**) 32 hpf, showing typical ciliated organs. (**G**) Left: preparation procedure of flat mount. Right: diagram of Kupffer's vesicle and floor plate in flat-mounted embryos.

DOI: https://doi.org/10.7554/eLife.36979.003

The following figure supplement is available for figure 1:

*Figure 1 continued on next page*

*Figure 1 continued*

**Figure supplement 1.** PIH gene mutants were established by genome-editing.

DOI: https://doi.org/10.7554/eLife.36979.004

spermatozoa showed a slight reduction of beating frequency (*Figure 2G*), and propagation of flagellar bending was disturbed, as slopes of shear angle curves changed between traces (*Figure 2B'*; asterisk). In *twister*[-/-], almost all spermatozoa were immotile, but a few were found to be motile with decreased beating frequencies and severely disturbed waveforms (*Figure 2E*). In *pih1d2*[-/-] and *ktu*[-/-], a significant difference was not observed in either beating frequencies or waveforms (*Figure 2C,D*). We suspected functional compensation of these two genes, and thus generated double mutants of *pih1d2*[-/-];*ktu*[-/-]. Double mutant spermatozoa exhibited abnormal waveforms; motile in the proximal half, while immotile in the distal half (*Figure 2F*). In the proximal region, beating frequency was about twice as high as that of wild type (*Figure 2G*). This could be caused by the reduction of sliding distance of DMTs, rather than the change of the sliding velocity of DMTs, because the slopes of shear angle curves was decreased (*Figure 2F'*; dotted line), indicating that the bending of the proximal flagella was smaller than that of wild type. We also analyzed the length of sperm flagella, but did not find any significant differences between wild type and PIH gene mutants (*Figure 2—figure supplement 1C*).

We then examined the motility of free swimming spermatozoa by CASA (computer-assisted sperm analysis) modified for zebrafish (*Wilson-Leedy and Ingermann, 2007*). In this analysis, traced paths of swimming sperm heads were used to calculate sperm motility (*Figure 2—figure supplement 1B*; *Video 2*). In all mutants, the ratios of motile (locomotive) spermatozoa were significantly decreased, and no sperm in *twister*[-/-] showed significant locomotion (*Figure 2H*). For motile spermatozoa, swimming velocity did not change in *pih1d1*[-/-], *pih1d2*[-/-], and *ktu*[-/-], but significantly decreased in *pih1d2*[-/-];*ktu*[-/-] (*Figure 2I*). The beating frequencies of sperm heads were decreased in *pih1d1*[-/-], but increased in *pih1d2*[-/-];*ktu*[-/-] (*Figure 2J*), which is consistent with *Figure 2G*.

## Cryo-ET revealed native ultrastructure of zebrafish sperm axoneme

To observe the ultrastructure of zebrafish axoneme, we applied cryo-ET to zebrafish spermatozoa. The axoneme of zebrafish sperm had the characteristic 9 + 2 arrangement of DMTs surrounding central-pair microtubules (*Figure 3—figure supplement 1A*). To analyze the structure of DMTs in more detail, subtomographic averaging was applied using the 96 nm repeat of DMTs assuming nine-fold rotational symmetry of the axoneme, since we did not detect any obvious heterogeneity of nine DMTs in zebrafish sperm unlike *Chlamydomonas* flagella and sea urchin sperm (*Hoops and Witman, 1983*; *Bui et al., 2012*; *Lin et al., 2012*). The averaged structure of zebrafish DMT exhibited overall similarity to that of other organisms (*Figure 3C, D*). Thus, based on the well-studied structure of the *Chlamydomonas* axoneme (*Bui et al., 2012*), we assigned the structures of OADs, seven types of IADs, radial spokes (RSs), and nexin-dynein regulatory complex (N-DRC) in the zebrafish axoneme (*Figure 3A,B*; *Video 3*).

To address the evolutionary conservation and diversity of cilia/flagella, we compared the ultrastructure of zebrafish axoneme to that of *Chlamydomonas* and human axonemes in more detail. Compared with *Chlamydomonas*, zebrafish axoneme does not have OAD α-HCs, but has longer RS3 (*Figure 3A,D*). A linker between N-DRC and OAD is not observed in the zebrafish axoneme unlike *Chlamydomonas* (*Figure 3D*; red arrowhead). These features of the zebrafish axoneme are also found in human respiratory cilia (*Figure 3C*; *Lin et al., 2014*). The same features were also reported in mouse

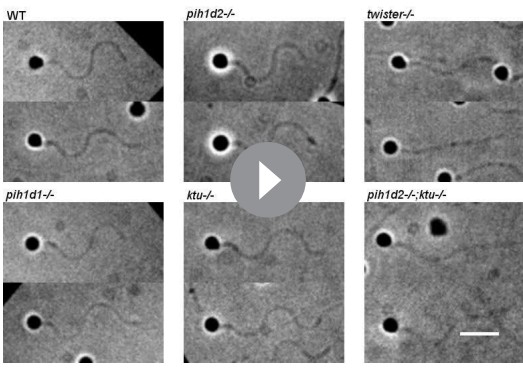

**Video 1.** Sperm waveforms of WT and PIH gene mutants. For each strain, two spermatozoa whose heads were attached to the coverslip are shown. Movies were filmed by a high-speed camera at 1000 fps and played at 30 fps. Scale bar: 10 μm.

DOI: https://doi.org/10.7554/eLife.36979.005

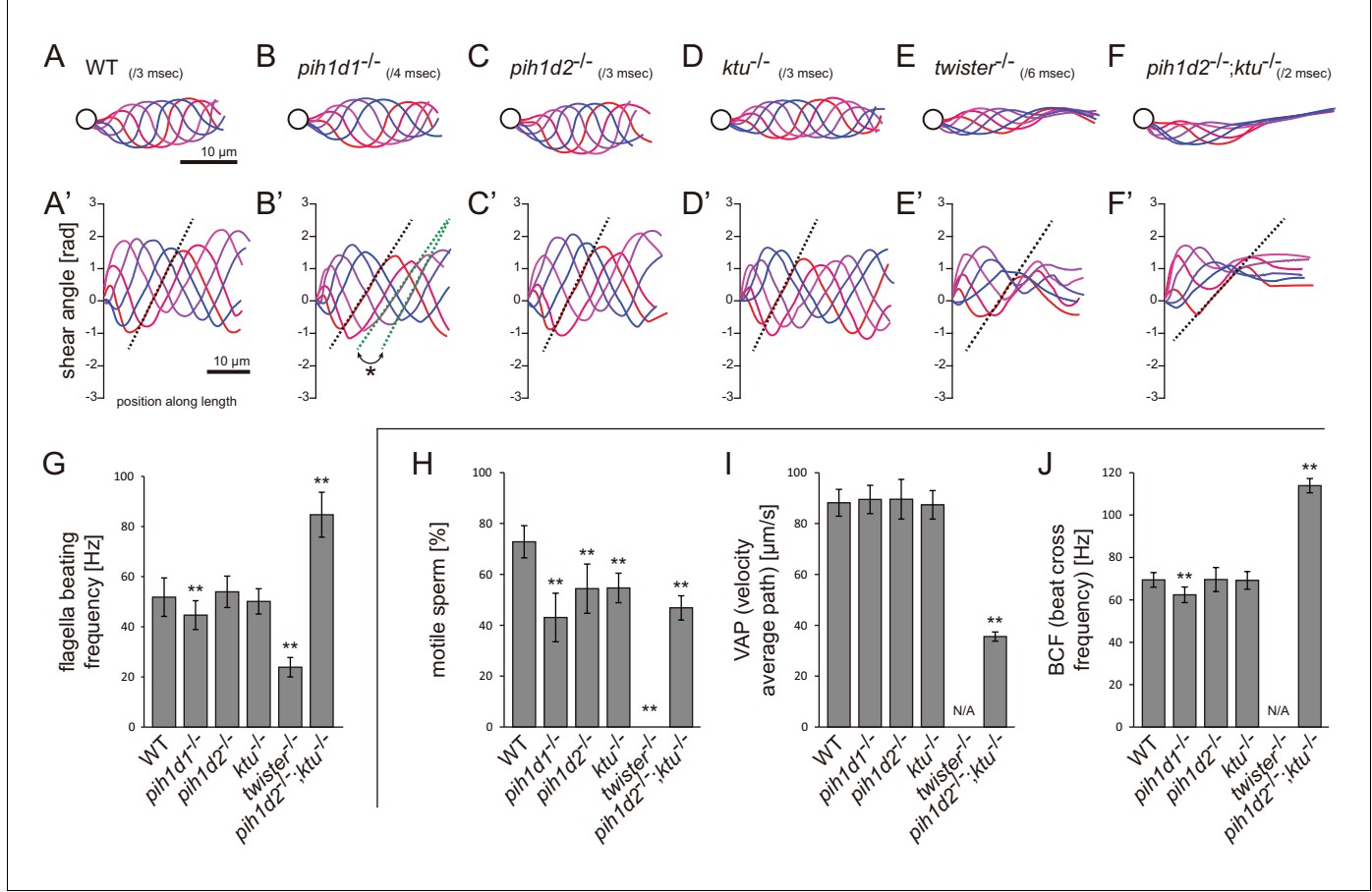

**Figure 2.** Mutations of PIH genes caused abnormal sperm motilities. (A–G) Waveforms and the beating frequencies of motile sperms were analyzed using a high-speed camera at 1000 fps (frames per second). (A–F) Sperm waveforms were traced six times in one beating cycle and overlaid. Intervals are as follows: WT, 3 ms; *pih1d1*[-/-], 4 ms; *pih1d2*[-/-], 3 ms; *ktu*[-/-], 3 ms; *twister*[-/-], 6 ms; and *pih1d2*[-/-];*ktu*[-/-], 2 ms. Traces are painted in a color gradient of from red (first traces) to blue (sixth traces). (A'–F') Shear angles of beating flagella were calculated from each trace of A-F. Slopes of dotted lines represent the size of flagellar bending. An asterisk with green dotted lines in B' shows unstable propagation of flagellar bending in *pih1d1*[-/-] sperm. (G) Beating frequencies of sperm flagella were measured by sperm kymographs. Kymographs of *pih1d2*[-/-];*ktu*[-/-] were obtained from the proximal region of sperm flagella, as distal flagella were immotile. Number of samples: WT, 23; *pih1d1*[-/-], 23; *pih1d2*[-/-], 26; *ktu*[-/-], 15; *twister*[-/-], 8; and *pih1d2*[-/-];*ktu*[-/-], 14. (H–J) Motilities of free-swimming spermatozoa were observed using a high-speed camera at 200 fps and analyzed by CASA modified for zebrafish. For each zebrafish line, more than 1200 spermatozoa were observed in total with sixteen technical replicates. Spermatozoa with less than 20 μm/s velocities were considered immotile. (H) Ratio of motile (locomotive) sperm. (I) Velocity of spermatozoa on averaged paths. Averaged paths were constructed by connecting the points of averaged sperm positions of coordinating 33 frames. (J) Frequencies at which sperm heads crossed their averaged paths. Bar graphs show mean ±SD. **: p-value<0.01 in Dunnett's test of each mutant against WT.

DOI: https://doi.org/10.7554/eLife.36979.006

The following source data and figure supplements are available for figure 2:

**Source data 1.** Data for *Figure 2G*, beating frequencies of sperm flagella.
DOI: https://doi.org/10.7554/eLife.36979.008

**Source data 2.** Data for *Figure 2H*, ratio of motile sperm.
DOI: https://doi.org/10.7554/eLife.36979.009

**Source data 3.** Data for *Figure 2I*, velocity of spermatozoa on averaged paths.
DOI: https://doi.org/10.7554/eLife.36979.010

**Source data 4.** Data for *Figure 2J*, frequencies at which sperm heads crossed their averaged paths.
DOI: https://doi.org/10.7554/eLife.36979.011

**Figure supplement 1.** Sperm motilities and flagellar length.
DOI: https://doi.org/10.7554/eLife.36979.007

**Figure supplement 1—source data 1.** Data for *Figure 2—figure supplement 1C*, length of sperm flagella.
DOI: https://doi.org/10.7554/eLife.36979.012

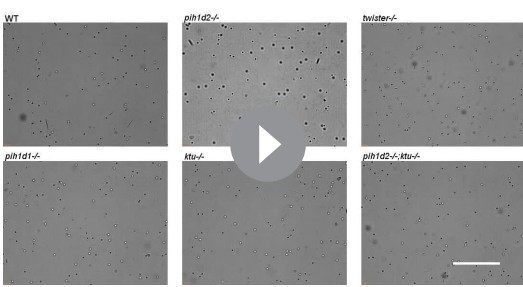

**Video 2.** Sperm motilities of WT and PIH gene mutants. Movies of free-swimming spermatozoa for CASA, filmed by a high-speed camera at 200 fps and played at 30 fps. Scale bar: 50 μm.
DOI: https://doi.org/10.7554/eLife.36979.013

respiratory cilia (*Ueno et al., 2012*) and found in sea urchin spermatozoa (*Lin et al., 2012*), indicating that these features are common among metazoans.

## Mutations of each PIH gene caused structural defects of different subtypes of axonemal dyneins

To gain structural insights into abnormal motility of mutant spermatozoa, we observed the structure of mutant axonemes by cryo-ET and subtomographic averaging. Compared to wild type, mutant axonemes exhibited structural defects of various types of axonemal dyneins (later summarized in *Figure 7A*). IAD c was missing in *pih1d1⁻ᐟ⁻* (*Figure 3F*; *Video 4)*, while no significant difference was observed in *pih1d2⁻ᐟ⁻* (*Figure 3G*; *Video 5*). In *ktu⁻ᐟ⁻*, smaller IAD c density was observed, suggesting that IAD c is partially missing in *ktu⁻ᐟ⁻* spermatozoa (*Figure 3H*; *Video 6*). In *twister⁻ᐟ⁻*, reflecting severe motility defects, OADs and IAD c were missing and smaller IAD g and d were observed (*Figure 3I*; *Video 7*). Intriguingly, in *pih1d2⁻ᐟ⁻;ktu⁻ᐟ⁻*, averaging of all DMT particles did not converge into one structure, thus tomograms were classified as follows. We noticed that out of nine tomograms of axonemes, four axonemes had OADs but five lacked OADs. Using this difference, we divided axonemes into two classes (+OAD and -OAD) and averaged, respectively. The +OAD class possessed a full set of axonemal dyneins, except for a smaller IAD c, like the *ktu⁻ᐟ⁻* axoneme (*Figure 3J*; *Video 8*). By contrast, the -OAD class lost not only OADs, but also IAD b, c, and e (*Figure 3K*; *Video 9*). However, note that the –OAD class showed faint densities of these IADs in the subtomographic slice (*Figure 3—figure supplement 1C*), which suggests that IAD b, c, and e were retained partially in the –OAD class axonemes. Although we found structural defects of axonemal dyneins, no significant defect was observed in other DMT structures, such as RSs, in all mutants we examined.

To correlate the structural defects of mutants with biochemical data, we performed immunoblot analysis of axonemal dynein components (*Figure 4A*). We made specific antibodies against zebrafish Dnah8 (OAD γ-HC) and Dnah2 (IAD f β-HC). Dnai1 is a component of OADs and is also known as IC1. Dnali1 is the orthologue of *Chlamydomonas* p28, which is the subunit of three types of IADs: IAD a, c, and d (*Piperno et al., 1990*; *Hom et al., 2011*). Consistent with the above structural analysis, Dnah8 and Dnai1 were missing from the axoneme of *twister⁻ᐟ⁻* (*Figure 4A*; asterisks). In *pih1d2⁻ᐟ⁻;ktu⁻ᐟ⁻*, the amount of Dnah8 and Dnai1 was decreased, possibly reflecting the presence of the two types of DMT structures (+OAD and -OAD). Dnah2 was not affected in any mutants, and so was the case of IAD f in the structural analysis. Dnali1 was slightly decreased in *pih1d1⁻ᐟ⁻*, *ktu⁻ᐟ⁻*, and *pih1d2⁻ᐟ⁻;ktu⁻ᐟ⁻* (*Figure 4A*; filled circles), confirming the loss of IAD c (one of three IADs containing p28 in *Chlamydomonas*) in these mutants. In *twister⁻ᐟ⁻*, the structural analysis revealed the loss of IAD c and d (two of three IADs containing p28), and the amount of Dnali1 was strongly reduced. Interestingly, shifted bands of Dnai1 were observed in *pih1d1⁻ᐟ⁻* and *pih1d2⁻ᐟ⁻* (*Figure 4A*; open circles), indicating abnormal construction of OADs in these mutants. However, the structure of OADs in these mutants appeared normal as far as our structural analysis showed at the current resolution. Taken together, all biochemical results are largely consistent with our structural data.

From these results, we conclude that all PIH proteins are responsible for the assembly of specific subtypes of axonemal dyneins. Together

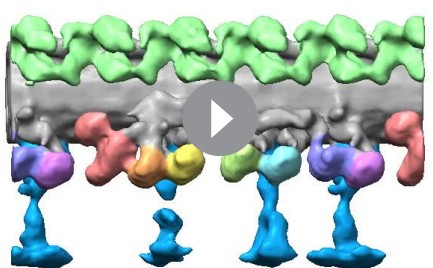

**Video 3.** 3D structure of DMT in WT sperm
DOI: https://doi.org/10.7554/eLife.36979.014

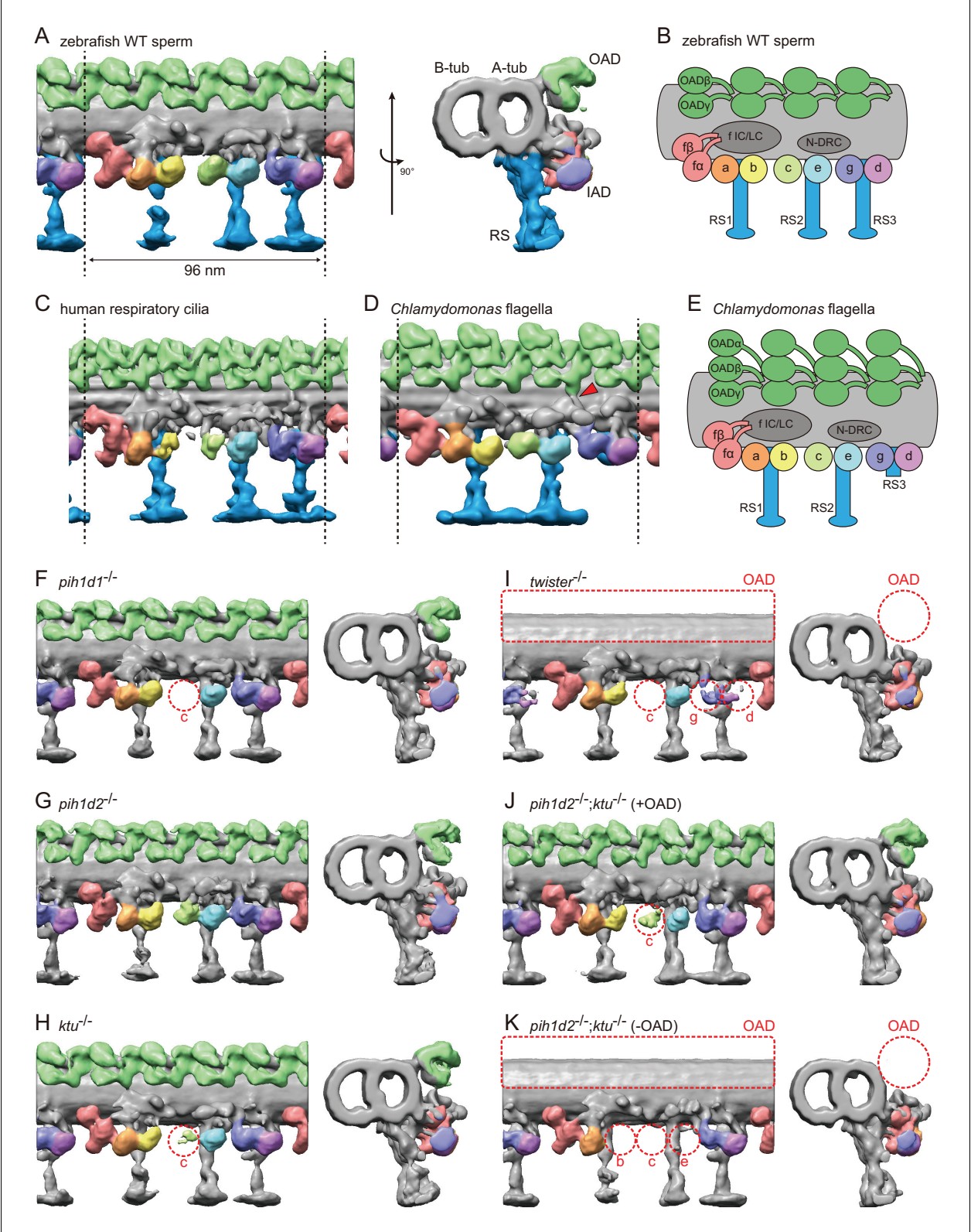

**Figure 3.** Cryo-ET revealed structural defects of axonemal dyneins in mutant spermatozoa. (**A**) DMT structure of native zebrafish sperm. Left: side view. Right: base-to-tip view. A-tub and B-tub: A- and B-tubule of DMT, respectively. (**B**) Diagram of the DMT structure of zebrafish sperm. f IC/LC means IAD f intermediate chain and light chain complex. (**C**) DMT structure of human respiratory cilia (EMD-5950; *Lin et al., 2014*). (**D**) DMT structure of *Chlamydomonas* flagella (DMT 2–8 averaged; EMD-2132; *Bui et al., 2012*). Red arrow indicates a linker between N-DRC and OAD. (**E**) Diagram of the
*Figure 3 continued on next page*

*Figure 3 continued*

DMT structure of *Chlamydomonas* flagella. (F–K) DMT structures of PIH gene mutant spermatozoa. For *pih1d2⁻/⁻;ktu⁻/⁻*, J (+OAD) and K (-OAD) represent subtomograms of axonemes with or without OADs, respectively. Red circles indicate the defects of axonemal dyneins. Green, OADs; red, IAD f; orange, IAD a; yellow, IAD b; light-green, IAD c; cyan, IAD e; indigo, IAD g; violet, IAD d; blue, RSs.

DOI: https://doi.org/10.7554/eLife.36979.015

The following figure supplement is available for figure 3:

**Figure supplement 1.** Tomographic slices and Fourier shell correlations of the averaged subtomograms.

DOI: https://doi.org/10.7554/eLife.36979.016

with their specific cytoplasmic localizations, we identified all vertebrate PIH proteins (not only Ktu and Twister, but also Pih1d1 and Pih1d2) as DNAAFs. Abnormal sperm motility observed in *pih1d1⁻/⁻*, *twister⁻/⁻* and *pih1d2⁻/⁻;ktu⁻/⁻* can be explained by the loss of specific subtypes of axonemal dyneins. On the other hand, spermatozoa of *pih1d2⁻/⁻* and *ktu⁻/⁻* appeared to have normal motility, although the structural or biochemical analyses revealed abnormal axonemal dyneins in these mutants. A likely explanation for this discrepancy is that the defects of axonemal dyneins in *pih1d2⁻/⁻* or *ktu⁻/⁻* spermatozoa are so subtle that other normal axonemal dyneins can compensate their loss of function. However, it is worth noting that the affected axonemal dyneins were different between *pih1d2⁻/⁻* (OAD Dnai1) and *ktu⁻/⁻* (IAD c), which indicates distinct functions of Pih1d2 and Ktu, although functional compensation of these two genes was also revealed by *pih1d2⁻/⁻;ktu⁻/⁻*.

## *pih1d2⁻/⁻;ktu⁻/⁻* spermatozoa have different axonemal structures between proximal and distal regions

The two types of DMT structures in *pih1d2⁻/⁻;ktu⁻/⁻*, (*Figure 3J,K*;+OAD and -OAD classes) led us to examine their distribution in the mutant axoneme. For this, we stained mutant spermatozoa with the anti-Dnah8 (OAD γ-HC) antibody (*Figure 4B*). In wild type, Dnah8 was localized along the entire length of the flagellum. However, in *pih1d2⁻/⁻;ktu⁻/⁻*, Dnah8 was consistently absent in the distal region, while it remained in the proximal (*Figure 4B*; white arrowhead). Thus, the +OAD class structure was localized in the proximal region, while the -OAD class was in the distal. We also analyzed the localization of Dnah8 in other mutants. Consistent with our structural analysis, in *pih1d1⁻/⁻*, *pih1d2⁻/⁻*, and *ktu⁻/⁻* spermatozoa, Dnah8 was normally distributed along the entire length of their flagella, while *twister⁻/⁻* spermatozoa completely lost Dnah8.

Different structural defects of IADs were also observed between the +OAD and -OAD classes. To assess the distribution of IADs in *pih1d2⁻/⁻;ktu⁻/⁻*, we analyzed the structure of proximal and distal axoneme directly. Among many cryo-prepared *pih1d2⁻/⁻;ktu⁻/⁻* axonemes, we found one axoneme suitable for observing both proximal and distal regions by cryo-ET (*Figure 4—figure supplement 1A*). Although the obtained subtomograms are noisy due to a smaller number of averaged particles, the DMT structure of a proximal subtomogram possessed OADs and the densities of all IADs, consistent with the structure of +OAD class (*Figure 4—figure supplement 1B,D*). On the other hand, the DMT structure of a distal subtomogram lost OADs, IAD b, c, and e, which corresponds to -OAD class (*Figure 4—figure supplement 1C,E*). Therefore, the distribution of not only OADs but also IADs is different between the proximal and distal regions in *pih1d2⁻/⁻;ktu⁻/⁻* spermatozoa.

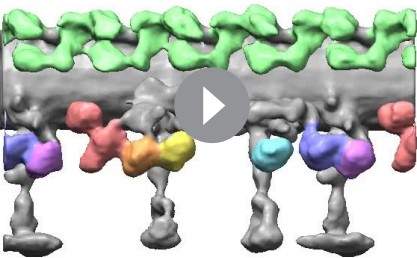

**Video 4.** 3D structure of DMT in *pih1d1⁻/⁻* sperm.
DOI: https://doi.org/10.7554/eLife.36979.017

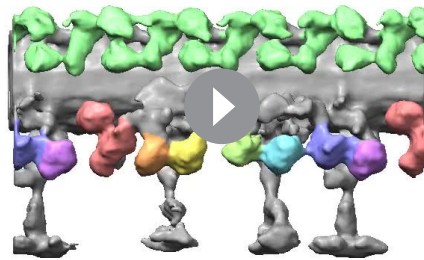

**Video 5.** 3D structure of DMT in *pih1d2⁻/⁻* sperm.
DOI: https://doi.org/10.7554/eLife.36979.018

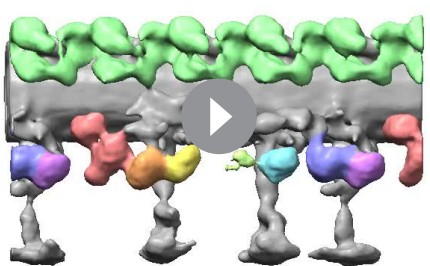

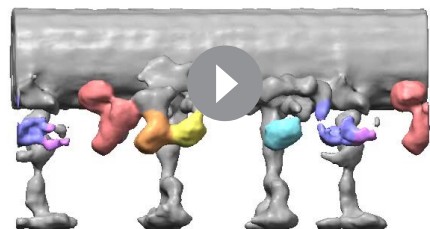

**Video 6.** 3D structure of DMT in *ktu*[-/-] sperm.
DOI: https://doi.org/10.7554/eLife.36979.019

**Video 7.** 3D structure of DMT in *twister*[-/-] sperm.
DOI: https://doi.org/10.7554/eLife.36979.020

## Kupffer's vesicle cilia showed different ciliary phenotypes from sperm flagella

In humans, organ-specific compositions of OAD HCs were reported between sperm flagella and respiratory cilia (Figure 7E; *Fliegauf et al., 2005*; *Dougherty et al., 2016*), and mouse *Pih1d3* was reported as a testis-specific gene (*Dong et al., 2014*). To assess organ-specific functions of zebrafish PIH proteins, we focused on a second ciliated organ, Kupffer's vesicle, which is orthologous to the mammalian embryonic node. In Kupffer's vesicle, epithelial cells project mono-cilia that have rotational motility to produce leftward fluid flow in the organ (*Figure 5A,B*). Like in the mouse node, this leftward flow is required for the determination of visceral asymmetry, and thus defects of Kupffer's vesicle cilia cause abnormal left-right patterning of the fish (*Essner et al., 2005*).

Mutations of each PIH gene caused abnormal motility of Kupffer's vesicle cilia. To describe ciliary motility, we categorized motion patterns into three classes: rotating, irregular, and immotile (*Figure 5C,D*; *Video 10*). Rotational frequencies were measured from rotating class cilia (*Figure 5E*). The resulting left-right patterning of embryos was assessed by observing the direction of heart looping (normally rightward; *Figure 5F,G*). In *pih1d1*[-/-], rotational frequencies of cilia were significantly reduced, but almost all cilia were motile and the ratio of heart-looping reversal was not largely affected. By contrast, in *twister*[-/-] and *pih1d2*[-/-];*ktu*[-/-], all cilia were immotile, leading to complete randomization of their left-right patterning. In *pih1d2*[-/-] and *ktu*[-/-], the proportions of rotating class cilia were decreased to ~40% and~15%, respectively, with reduced rotational frequencies, resulting in significant levels of heart-looping defects (*Figure 5D*). Regarding the structure and localization of axonemal dyneins, due to technical difficulties, we were unable to apply cryo-ET and immunohistochemistry with anti-dynein antibodies to the axonemes of Kupffer's vesicle cilia.

Together with sperm analyses, we conclude that all PIH genes of zebrafish are essential for normal motility of both sperm flagella and Kupffer's vesicle cilia. Intriguingly, however, in *pih1d2*[-/-] and *ktu*[-/-], only Kupffer's vesicle cilia showed motility defects, while sperm flagella beat normally, indicating that Pih1d2 and Ktu have organ-specific functions.

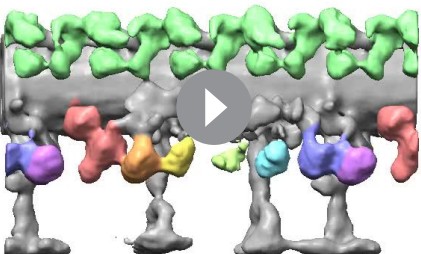

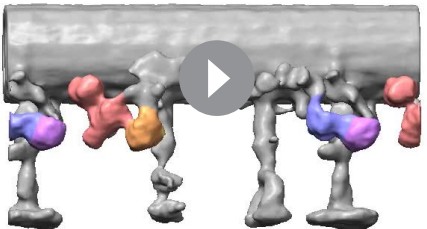

**Video 8.** 3D structure of DMT (+OAD) in *pih1d2*[-/-];*ktu*[-/-] sperm.
DOI: https://doi.org/10.7554/eLife.36979.021

**Video 9.** 3D structure of DMT (-OAD) in *pih1d2*[-/-];*ktu*[-/-] sperm.
DOI: https://doi.org/10.7554/eLife.36979.022

## Testis and Kupffer's vesicle showed different expression patterns of DNAH genes

The organ-specific phenotypes of PIH mutants could reflect organ-specific compositions of axonemal dyneins. To address this, we performed whole-mount in situ hybridization of various DNAH genes. Zebrafish have three OAD β-HC genes: *dnah9*, *dnah9l*, and *dnah11*, and two OAD γ-HC genes: *dnah5* and *dnah8*. As for IAD, *dnah2* is an IAD f β-HC gene, and *dnah3* and *dnah7l* are other IAD HC genes. The gene correspondence of dynein heavy chains among zebrafish, human, and *Chlamydomonas* are summarized in *Table 1*, based on the comprehensive analysis of dynein phylogeny by *Kollmar (2016)*.

Zebrafish embryos and testes showed distinct expression patterns of DNAH genes (*Figure 6A*). When comparing Kupffer's vesicle and testis, *dnah11* expression was specifically detected in Kupffer's vesicle, while *dnah8* and *dnah3* were specifically detected in testis (*Figure 6B*). At the embryonic stages, *dnah9l* and *dnah8* were detected only in the otic vesicle and the pronephric duct, respectively, which also suggested specific combinations of DNAH genes in these organs. These results indicate that components of axonemal dyneins are indeed organ-specific. Intriguingly, however, Kupffer's vesicle and floor plate, whose cilia exhibit similar rotational motility (*Kramer-Zucker et al., 2005*), showed the same expression patterns of DNAH genes. It is likely that the same transcriptional regulation is required in Kupffer's vesicle and floor plate to construct similar types of cilia.

## Discussion

In the process of ciliary construction, the PIH protein family has been implicated in the preassembly of different subsets of axonemal dyneins, but thus far there has been insufficient evidence to support this idea. We performed the systematic analysis of all PIH proteins using zebrafish, and demonstrated that all of vertebrate PIH proteins (including two novel ciliary factors: Pih1d1 and Pih1d2) are required for cilia/flagella motions and the assembly of axonemal dyneins. Each PIH protein was responsible for the construction of different subsets of axonemal dyneins, suggesting the cytoplasmic assembly pathways for different axonemal dyneins through distinct PIH proteins.

### PIH proteins are required for the constructions of specific subsets of axonemal dyneins

Our cryo-ET and biochemical analyses revealed that the PIH proteins are required for the assemblies of specific subsets of axonemal dyneins (*Figure 7C*): Pih1d1 for OAD (Dnai1 construction) and IAD c; Pih1d2 and Ktu for OAD, IAD b, c, and e; and Twister for OAD, IAD c, g, and d. In *Chlamydomonas*, the mutation of KTU/PF13 affected the assembly of OAD and IAD c, while the mutation of MOT48 affected the assembly of OAD, IAD b, c, d, and e (*Yamamoto et al., 2010*). Although the affected subtypes of axonemal dyneins are not the same, the mutations of PIH genes resulted in the loss of specific subsets of axonemal dyneins in both organisms. Remarkably, OAD and IAD c are most sensitive to the mutations of various PIH genes in both zebrafish and *Chlamydomonas*. Consistent with this, *Dong et al. (2014)* suggested that the assembly of OAD proceeds in a stepwise manner mediated by different PIH proteins. This might also be the case for IAD c, although no experimental evidence has been obtained.

In our research, IAD a or f were not affected in any PIH gene mutants (*Figure 7C*), which was essentially the same as in *Chlamydomonas* mutants of KTU/PF13 and MOT48. This suggests that these axonemal dyneins are assembled independently of PIH proteins. Alternatively, multiple PIH proteins redundantly participate in their assembly. IAD a or f may not be constructed automatically, because a defect of DYX1C1 (a known DNAAF other than PIH family proteins; *Tarkar et al., 2013*) affects the normal assembly of all types of IADs in *Chlamydomonas* (*Yamamoto et al., 2017*). Double, triple or quadruple mutants of PIH genes will be needed to answer this question.

### Comparison of zebrafish PIH protein functions with other reports

The phenotypes of zebrafish PIH gene mutants are summarized in *Figure 7A,B*. Although PIH1D1 has been known to serve as a component of R2TP complex, which has various important cellular functions (*Kakihara and Houry, 2012*), the role of PIH1D1 in vertebrate development remains

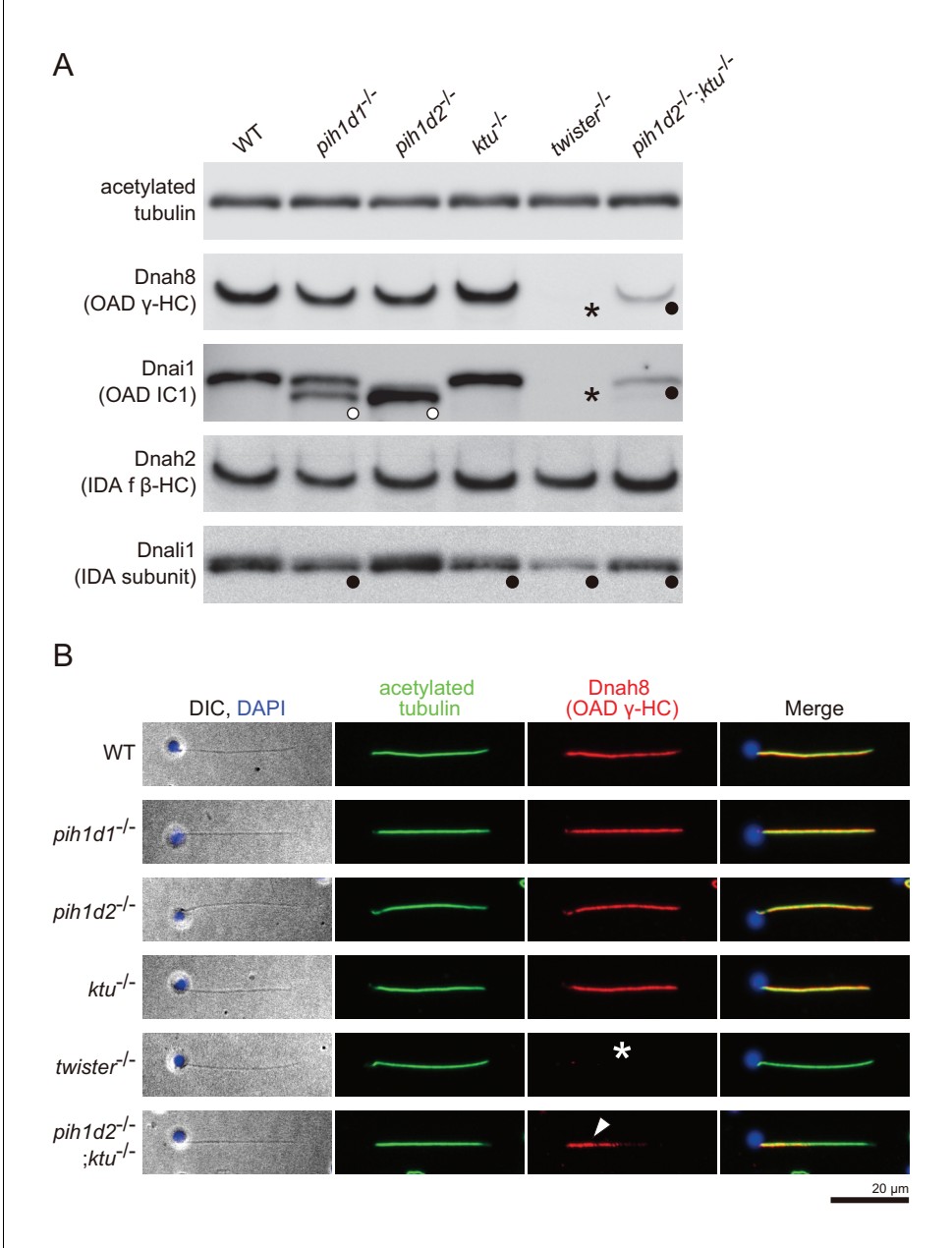

**Figure 4.** Immunoblot and immunofluorescence microscopy of axonemal dyneins. (**A**) Immunoblot of sperm axonemes. Asterisks, filled circles, and open circles indicate missing, decreased, and shifted bands, respectively. Acetylated tubulin: control. (**B**) Immunofluorescence microscopy of zebrafish spermatozoa. Dnah8 was localized along the entire length of sperm flagella in WT, *pih1d1*−/−, *pih1d2*−/−, *ktu*−/−. In *twister*−/−, Dnah8 was lost (white asterisk), while in *pih1d2*−/−;*ktu*−/−, Dnah8 was localized only in the proximal half of the flagella (white arrowhead).
DOI: https://doi.org/10.7554/eLife.36979.023

The following figure supplement is available for figure 4:

**Figure supplement 1.** Axonemal structures in *pih1d2*−/−;*ktu*−/− sperm.
DOI: https://doi.org/10.7554/eLife.36979.024

poorly understood. Zebrafish *pih1d1*−/− mutants were viable and exhibited only ciliary defects as far as we observed. Thus, Pih1d1 could be mostly redundant in cellular functions. In fact, yeast *PIH1*-deletion cells were also reported to be viable (*Gonzales et al., 2005*), like zebrafish *pih1d1*−/−.

Accumulated knowledge about DNAAFs has suggested the involvement of R2TP-like complex in the process of axonemal dynein assembly (*Li et al., 2017*). We identified Pih1d1 as a novel DNAAF,

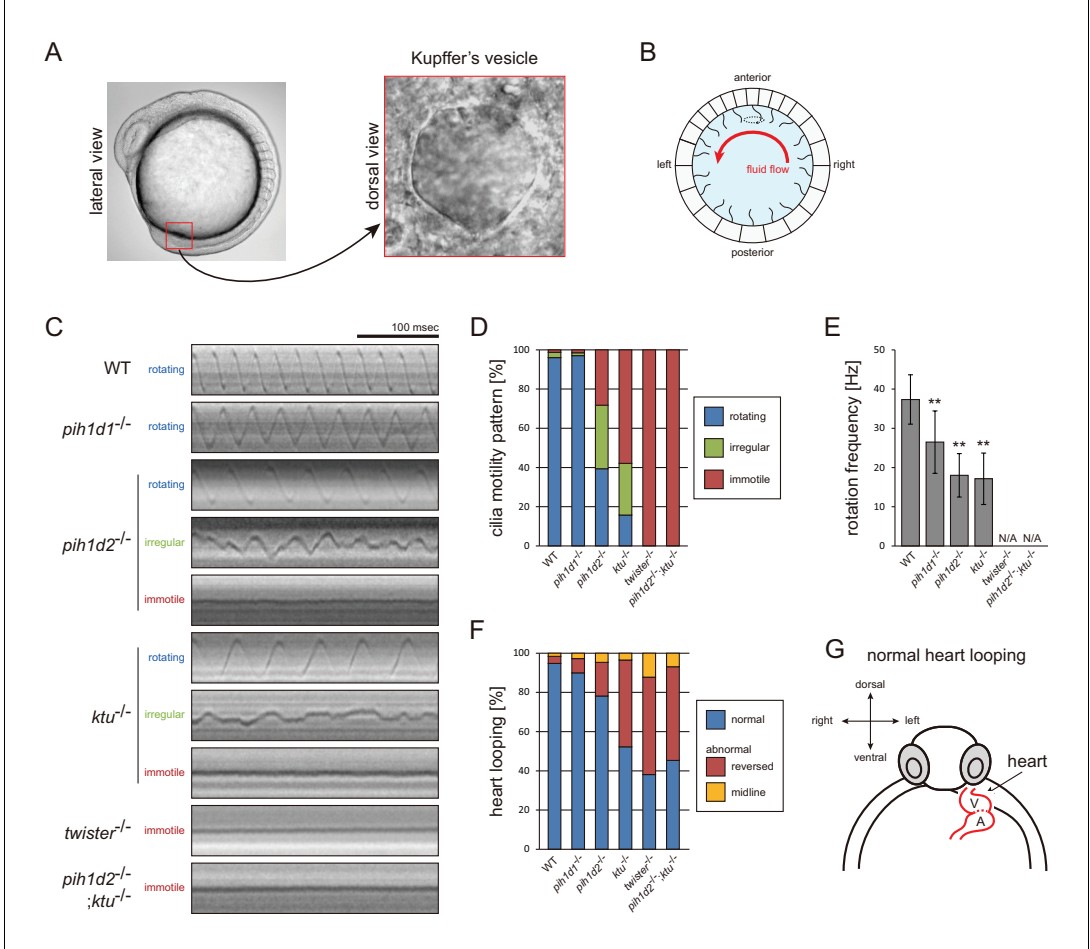

**Figure 5.** Mutations of PIH genes caused abnormal motilities of Kupffer's vesicle cilia. (**A**) Images of zebrafish embryo and Kupffer's vesicle. Red square in left-hand image indicates the position of the Kupffer's vesicle in a 12 hpf embryo. Right-hand image shows the dorsal view of the Kupffer's vesicle. (**B**) Diagram of zebrafish Kupffer's vesicle. Mono-cilia, which project from epithelial cells, have rotational motilities to produce left-ward fluid flow (red arrow) in the organ. (**C**) Typical kymographs of Kupffer's vesicle cilia in WT and PIH gene mutants. Patterns of the kymographs were categorized into three classes: rotating (blue), irregular (green), and immotile (red). (**D**) Ratios of each motility class. Number of observed cilia: WT, 76; *pih1d1*[-/-], 67; *pih1d2*[-/-], 198; *ktu*[-/-], 190; *twister*[-/-], 53; and *pih1d2*[-/-];*ktu*[-/-], 48. (**E**) Rotational frequencies of Kupffer's vesicle cilia. Bar graphs show mean ±SD. **: p-value<0.01 in Dunnett's test of each mutant against WT. (**F**) Heart looping of WT and mutant embryos at 30 hpf. Number of samples: WT, 247; *pih1d1*[-/-], 288; *pih1d2*[-/-], 275; *ktu*[-/-], 285; *twister*[-/-], 139; and *pih1d2*[-/-];*ktu*[-/-], 276. (**G**) Diagram represents normal heart looping of the 30 hpf embryo. V, ventricle; A, atrium.

DOI: https://doi.org/10.7554/eLife.36979.025

The following source data is available for figure 5:

**Source data 1.** Data for *Figure 5E*, rotational frequencies of Kupffer's vesicle cilia.
DOI: https://doi.org/10.7554/eLife.36979.026

which strongly support this idea. Since KTU and PIH1D3 are also suggested to participate in R2TP-like complexes (*Tarkar et al., 2013*; *Olcese et al., 2017*), each of PIH proteins may serve as a component of R2TP-like complexes. Intriguingly, however, our expression analysis of PIH genes suggested that *pih1d2*, *ktu*, and *twister* have cilia-specific functions, while *pih1d1* has ubiquitous cellular functions in addition to ciliary function. Further analysis of binding partners of PIH proteins can provide us the mechanism of how PIH1D1 promote the assembly of various types of protein complexes and how distinct PIH proteins modulate the assembly of different types of axonemal dyneins.

It was surprising that the spermatozoa of zebrafish *ktu*[-/-] showed normal motility, since abnormal sperm motility was reported in both human *KTU/DNAAF2*[-/-] patients and medaka *ktu* mutants (*Omran et al., 2008*). This suggests that the function of Ktu and other PIH proteins in zebrafish could have diverged during evolution. Intriguingly, double mutants of *pih1d2*[-/-];*ktu*[-/-] showed phenotypes

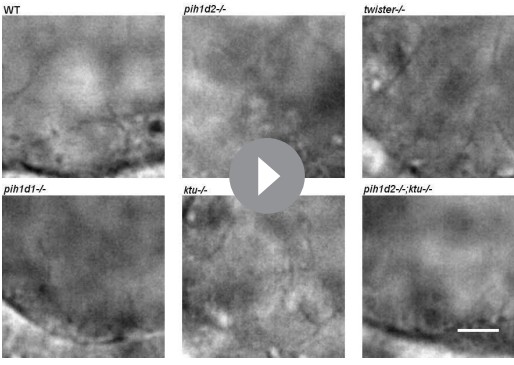

**Video 10.** Motilities of Kupffer's vesicle cilia of WT and PIH gene mutants. Typical movies of Kupffer's vesicle cilia, filmed by a high-speed camera at 1000 fps and played at 30 fps. Scale bar: 5 µm.
DOI: https://doi.org/10.7554/eLife.36979.027

similar to those of the medaka *ktu* mutant, in terms of complete loss of ciliary motility in Kupffer's vesicle (*Figure 5D*) and the expansion of pronephric ducts (*Figure 7—figure supplement 1F*). Furthermore, the waveform of *pih1d2⁻/⁻;ktu⁻/⁻* spermatozoa highly resembles that of medaka *ktu* mutant spermatozoa (bends do not propagate to the tip of the sperm tail; *Omran et al., 2008*). The function of medaka Ktu is thus partially shared by Ktu and Pih1d2 in zebrafish. Such functional divergence of PIH proteins was also reported in human and mouse; human *PIH1D3* has functions in various ciliated organs (*Paff et al., 2017*; *Olcese et al., 2017*), while mouse *Pih1d3* is a testis-specific gene (*Dong et al., 2014*). Twister2 (paralogue of mouse *Pih1d3*) is expressed in various ciliary organs including testis in mice but is not able to rescue the loss of *Pih1d3* in testis. Therefore, PIH proteins tend to be functionally diverse and sometimes interchangeable, even though they stay in the category of DNAAFs.

## Sperm axoneme of *pih1d2⁻/⁻;ktu⁻/⁻* exhibited distal-specific loss of axonemal dyneins

In *pih1d2⁻/⁻;ktu⁻/⁻* spermatozoa, OAD, IAD b, c, and e were missing only from the distal region of the flagella. One possible explanation for this phenotype is that the lack of both PIH1d2 and Ktu causes decreased efficiency of axonemal dynein assembly, leading to a shortage of dyneins to be loaded in the distal axoneme. Actually, the axoneme is known to continue elongating by adding flagellar components to its distal end during ciliogenesis (*Johnson and Rosenbaum, 1992*), and in mature

**Table 1.** Correspondence of DNAH genes/proteins among zebrafish, human, and *Chlamydomonas*.

| | Gene/protein names | | |
| --- | --- | --- | --- |
| | **Zebrafish** | **Human** | ***Chlamydomonas*** |
| OAD HCs | | | OAD α (DHC13) |
| | Dnah9 | DNAH9 | OAD β (DHC14) |
| | Dnah11 | DNAH11 | |
| | Dnah9l | | |
| | | DNAH17 | |
| | Dnah5 | DNAH5 | OAD γ (DHC15) |
| | Dnah8 (Dnah5l) | | |
| | | DNAH8 | |
| IAD f HCs | Dnah10 | DNAH10 | IAD fα (DHC1) |
| | Dnah2 | DNAH2 | IAD fβ (DHC10) |
| IAD HCs | Dnah3 | DNAH3 | IAD a (DHC6) |
| | Dnah7 | DNAH7 | IAD b (DHC5) IAD c (DHC9) |
| | Dnah12 | DNAH12 | IAD e (DHC8) minor var. DHC11 minor var. DHC4 |
| | Dnah1 | DNAH1 | IAD d (DHC2) |
| | Dnah7l (Dnah6) | DNAH6 | IAD g (DHC7) minor var. DHC3 |
| | | DNAH14 | |

DOI: https://doi.org/10.7554/eLife.36979.028

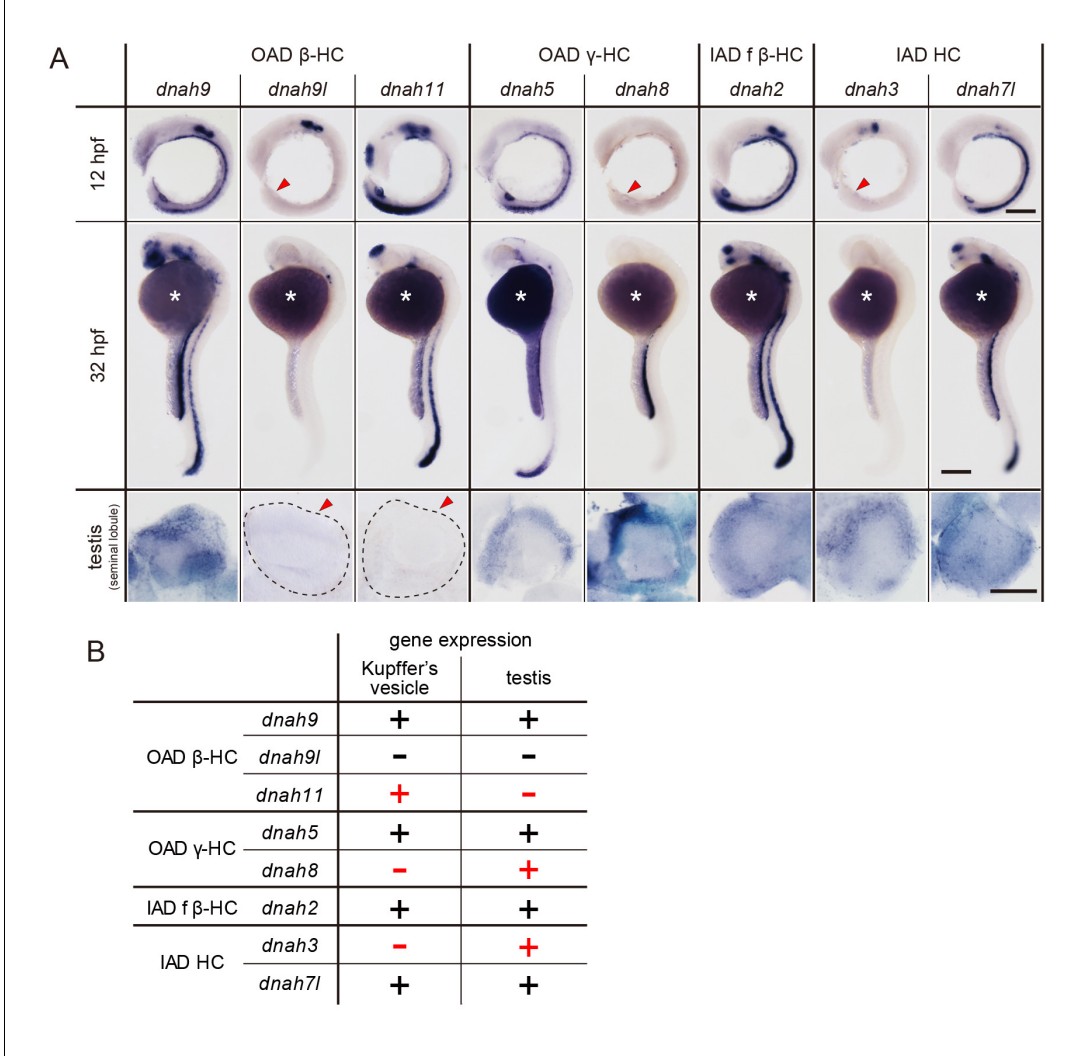

**Figure 6.** DNAH genes showed distinct expression patterns in zebrafish embryos and testis. (A) Whole-mount in situ hybridization of DNAH genes with embryos (12 and 32 hpf) and dissected testis (seminal lobule). For embryos, lateral views are shown. Yolk of 12 hpf embryos was removed before observation to show Kupffer's vesicle clearly. Red arrowheads indicate Kupffer's vesicles or testes in which DNAH gene expressions were not detected. White asterisks: non-specific staining of yolk. Scale bars: 200 µm for embryos; 100 µm for testes. (B) Comparison of DNAH gene expression between Kupffer's vesicle and testis. +, expressed; -, expression not detected. Red indicates when DNAH gene expression was difference between the two organs.

DOI: https://doi.org/10.7554/eLife.36979.029

spermatozoa, the transport of flagellar components is highly improbable, because IFT components disappear as spermatozoa mature (*San Agustin et al., 2015*). Alternatively, the distal and proximal region of zebrafish sperm could differ in the composition of axonemal dyneins. Indeed, human respiratory cilia are known to have two types of OADs, that is DNAH11/DNAH5-containing OADs in the proximal and DNAH9/DNAH5-containing OADs in the distal parts (*Figure 7E*; *Fliegauf et al., 2005*; *Dougherty et al., 2016*). Intriguingly, a mutation in the human KTU/DNAAF2 gene strongly affects the assembly of only distal OADs in respiratory cilia (*Omran et al., 2008*). However, at the moment, we do not have any evidence for the distal-specific dynein composition in zebrafish spermatozoa. Although testis showed the expression of two OAD γ-HC genes: *dnah5* and *dnah8*, Dnah8 is present along the entire length of sperm flagella (*Figure 4B*) and the distribution of Dnah5 is not known. As for OAD β-HC gene, only *dnah9* was detectable in testis (*Figure 6*). Further analyses with $pih1d2^{-/-}$; $ktu^{-/-}$ spermatozoa could shed light on the structural and functional difference between distal and proximal regions of vertebrate spermatozoa.

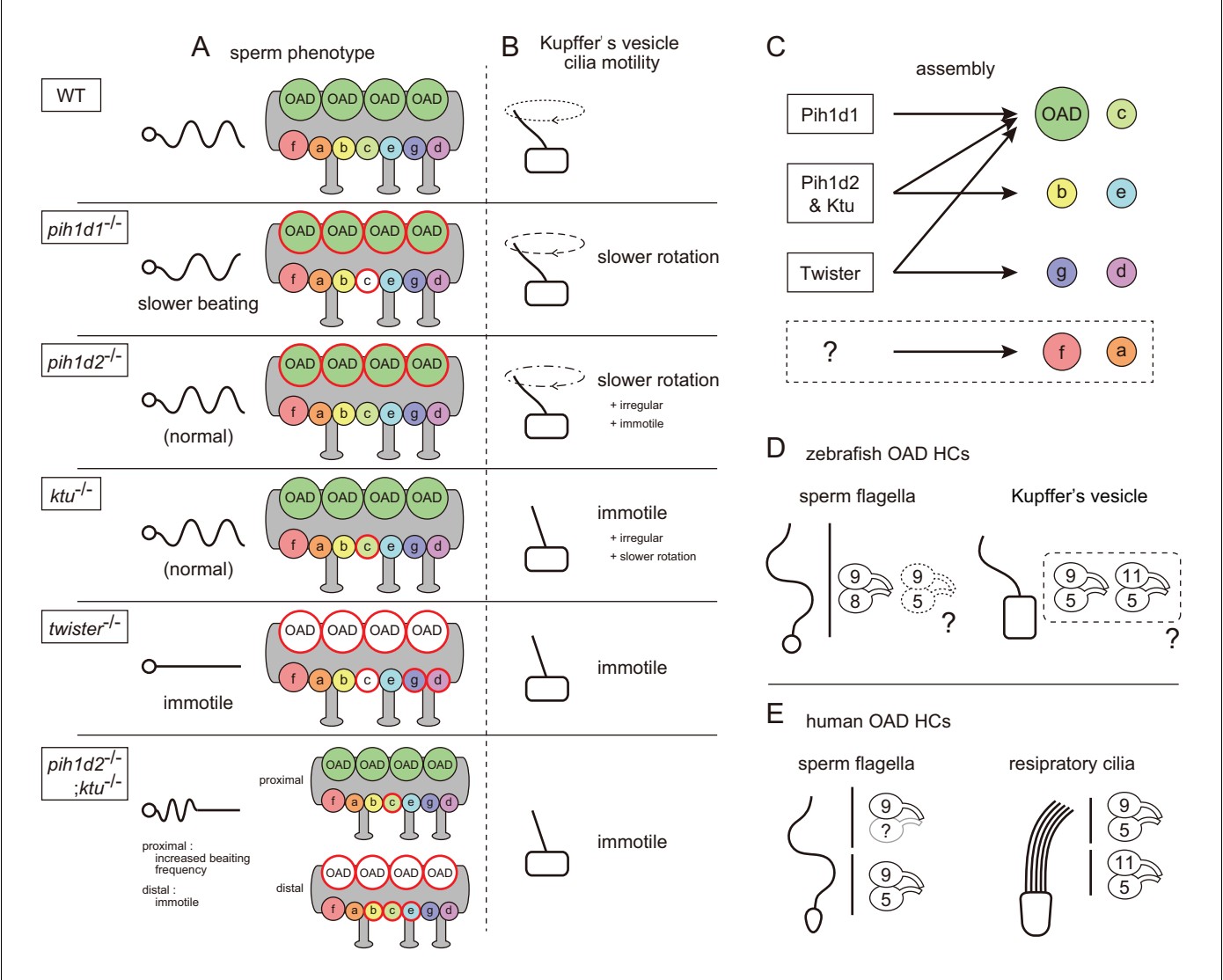

**Figure 7.** Mutant phenotypes revealed axonemal dyneins assembled by distinct PIH proteins in zebrafish. (A) Summary of the sperm motilities and the axonemal structures of WT and each PIH gene mutant. Abnormal axonemal dyneins are indicated by red circles filled with white (missing) or color (diminished or Dnai1 defects). (B) Summary of the motilities of Kupffer's vesicle cilia. (C) Proposed model of the axonemal dynein assembly by distinct PIH proteins. Pih1d1, Pih1d2 and Ktu, and Twister are required for the assembly of specific subsets of axonemal dyneins. Factors responsible for the construction of IAD a or IAD f were not revealed in our research. (D) Possible combinations of OAD HCs in zebrafish sperm flagella and Kupffer's vesicle cilia, suggested by the expression analysis of DNAH genes. Although testis showed expression of both γ-HC genes (*dnah5* and *dnah8*), Dnah8 were present along the entire length of sperm flagella, while the distribution of Dnah5 is not known. (E) Possible combinations of OAD HCs in human sperm flagella and respiratory cilia. Localizations of DNAH9 and DNAH5 in human spermatozoa were reported by *Fliegauf et al. (2005)*. Localizations of DNAH9, DNAH11 and DNAH5 in human respiratory cilia were reported by *Dougherty et al. (2016)*.
DOI: https://doi.org/10.7554/eLife.36979.030

The following figure supplement is available for figure 7:

**Figure supplement 1.** Expansion of pronephric duct was observed in *twister*[-/-] and *pih1d2*[-/-];*ktu*[-/-].
DOI: https://doi.org/10.7554/eLife.36979.031

## Different compositions of axonemal dyneins between testis and Kupffer's vesicle

Expression analysis of DNAH genes suggested that the composition of axonemal dyneins differed between sperm flagella and Kupffer's vesicle cilia. In sperm flagella, Dnah9/Dnah8-containing OADs could be majority as discussed above, while in Kupffer's vesicle cilia, the axonemes seem to be

constituted of Dnah9/Dnah5-containing OADs and/or Dnah11/Dnah5-containing OADs (**Figure 7D**). We also found that *dnah3* (IAD HC gene) are differently expressed between testis and Kupffer's vesicle (**Figure 6**). Given that all PIH proteins are expressed in the two organs, phenotypic differences between *pih1d2*[-/-] and *ktu*[-/-] can be accounted for by the different compositions of axonemal dyneins. It is tempting to speculate that the dynein compositions vary depending on the pattern of ciliary beating, as cilia and flagella of the two organs exhibit the different mode of movement, planar oscillation for sperm flagella and rotation for Kupffer's vesicle cilia.

# Materials and methods

## Key resources table

| Reagent type (species) or resource | Designation | Source or reference | Identifiers | Additional information |
|---|---|---|---|---|
| Gene (*Danio rerio*) | pih1d1 | NA | ZFIN:ZDB-GENE-050309–147 | |
| Gene (*D. rerio*) | pih1d2 | NA | ZFIN:ZDB-GENE-041212–54 | |
| Gene (*D. rerio*) | ktu | NA | ZFIN:ZDB-GENE-050809–128 | |
| Gene (*D. rerio*) | twister | NA | ZFIN:ZDB-GENE-040722–2 | |
| Antibody | anti-Pih1d1 | This paper | | Rabbit polyclonal; against full length |
| Antibody | anti-Pih1d2 | This paper | | Rabbit polyclonal; against full length |
| Antibody | anti-Ktu | This paper | | Rabbit polyclonal; against full length |
| Antibody | anti-Twister | This paper | | Rabbit polyclonal; against full length |
| Antibody | anti-Dnah8 | This paper | | Rabbit polyclonal; against aa 895–1402 |
| Antibody | anti-Dnah2 | This paper | | Rabbit polyclonal; against aa 802–1378 |
| Antibody | anti-Dnai1 | GeneTex | GeneTex:GTX109719 | |
| Antibody | anti-Dnali1 | PMID: 7579689 | | anti-p28 antibody |
| Antibody | anti-α-tubulin (mouse monoclonal) | Sigma | Sigma:T9026 | |
| Antibody | anti-acetylated tubulin (mouse monoclonal) | Sigma | Sigma:T6793 | |
| Software, algorithm | CASA modified for zebrafish | PMID:17137620 | | |
| Software, algorithm | IMOD | PMID:8742726 | | |
| Software, algorithm | PEET | PMID:16917055 | | |
| Software, algorithm | UCSF Chimera | PMID:15264254 | | |

## Zebrafish maintenance

Zebrafish were maintained at 28.5°C on a 13.5/10.5 hr light/dark cycle. Embryos and larvae were raised at the same temperature in 1/3 Ringer's solution (39 mM NaCl, 0.97 mM KCl, 1.8 mM CaCl$_2$, and 1.7 mM HEPES, pH 7.2). Developmental stages of embryos and larvae are described according to hpf at 28.5°C and the morphological criteria by *Kimmel et al. (1995)*. For embryos used in whole-mount in situ hybridization, 200 µM 1-phenyl-2-thiourea was added to 1/3 Ringer's solution to delay pigmentation.

## Identification of zebrafish PIH proteins

PIH1 domain is registered as PF08190 in the Pfam database. To find all PIH proteins in the zebrafish genome, BLASTp search was performed with the consensus sequence of the PIH1 domain (https://www.ncbi.nlm.nih.gov/Structure/cdd/cddsrv.cgi?uid=pfam08190). Four proteins were identified as a match: Pih1d1 (NP_001153400.1, E value = 2.66e-27), Pih1d2 (NP_001008629.1, E value = 2.08e-9), Ktu (NP_001028272.1, E value = 2.37e-31), and Twister (also known as Pih1d3; NP_001002309.1, E value = 4.16e-5). Paralogues of each match were also checked, since teleost fish are known to have undergone an additional genome duplication. BLASTn and tBLASTn search were performed using each PIH sequence as a query; however, only the proteins containing the query sequence were a hit in E value <10. Therefore, zebrafish have four PIH proteins: Pih1d1, Pih1d2, Ktu, and Twister.

## Zebrafish genome-editing

Zebrafish genome-editing was performed according to previous reports of TALEN (*Bedell et al., 2012*) or CRISPR/Cas9 (*Gagnon et al., 2014*). Target sites of our genome-editing are as follows: *pih1d1* (TALEN left), GTTGAACACGAGCAGAAACAA; *pih1d1* (TALEN right), TGAAGCAGAAGTTG TTGGTA; *pih1d2* (TALEN left), TACAGGAGCTTCATTCAG; *pih1d2* (TALEN right), TGAGTGAAAC TCGGCTCCC; *ktu* (CRIPSR gRNA), GGAGATCCGGCCACAGCTGG; *twister* (CRISPR gRNA), GGA TAATGATGAGGAAGAAG. Genomic DNA was extracted from the developing embryos and target loci were amplified to check the mutations by sanger-sequencing. After identifying founder fish, each mutant line underwent back-cross twice to remove the effect of possible off-target mutations.

## Sperm treatment

Zebrafish sperm was expelled by gently squeezing the sides of the fish, and collected in Hank's buffer (137 mM NaCl, 5.4 mM KCl, 0.25 mM $Na_2HPO_4$, 0.44 mM $KH_2PO_4$, 1.3 mM $CaCl_2$, 1.0 mM $MgSO_4$, and 4.2 mM $NaHCO_3$). For the fractionation of sperm head and flagella, spermatozoa were passed through a 26-gauge needle 20 times in Hank's buffer with 2 mg/ml BSA, and the separated heads and flagella were collected by centrifugation (head: 400 *g*, 3 min; flagella: 9000 *g*, 3 min). For purification of sperm axonemes, sperm heads and membranes were removed by adding 2% Nonidet P-40 to Hank's buffer, and demembranated axonemes were collected by centrifugation (10,000 *g*, 3 min), then resuspended in HMDEKAc buffer (30 mM HEPES at pH 7.2, 5 mM $MgSO_4$, 1 mM dithiothreitol, 1 mM EGTA, and 50 mM $CH_3COOK$).

## Sperm motility analyses

To observe proper motility, spermatozoa were kept on wet ice until analyzed and used within 1 hr of sperm collection. Zebrafish spermatozoa were inactive in Hank's buffer, but were activated by adding abundant amount of 1/5 × Hank's buffer. Sperm motilities were observed under bright-field conditions using an inverted microscope (DMI6000B; Leica) and a high-speed camera (HAS-L1; Detect). For waveform analysis, spermatozoa whose heads were attached to the coverslip were selected and waveforms of flagella were filmed at 1000 fps. On the other hand, for CASA, 2 mg/ml of BSA was added to buffers to prevent sperm from attaching to the glass and free swimming spermatozoa were filmed at 200 fps. CASA modified for zebrafish was performed as previously reported (*Wilson-Leedy and Ingermann, 2007*). Spermatozoa were prepared on glass slides with 10 μm spacers (200A10; Kyodo giken chemical), and covered with coverslips to provide a consistent fluid depth. Eight independent experiments with two times of 1 s observations were performed to obtain 16 technical replicates of CASA.

## Cryo-preparation of zebrafish sperm axoneme

Purified sperm axoneme were incubated with anti-α-tubulin antibody (1:10000 dilution; T9026; Sigma-Aldrich) for 15 min at 4°C in HMDEKAc buffer, and then with anti-mouse antibody conjugated with 15 nm colloidal gold (final 1:50 dilution; EM.GMHL15; BBInternational) and 15 nm colloidal gold conjugated with BSA (final 1:5 dilution; 215.133; Aurion) were added. Holey carbon grids were glow discharged before use to make them hydrophilic. 5 μl of axoneme solution was loaded onto the grid, and then excess liquid was blotted away with filter paper to make a thin film of the solution. Immediately after, the grid was plunged into liquid ethane at −180°C for a rapid freeze of the solution. Blotting and freezing were automatically performed by an automated plunge-freezing device

(EM GP; Leica). Cryo-prepared grids were stored in liquid nitrogen until observation using the electron microscope.

## Cryo-image acquisition and image processing

Cryo-prepared grids were transferred into a transmission electron microscope (JEM-3100FEF; JEOL) with a high-tilt liquid nitrogen cryotransfer holder (914; Gatan), and kept at −180°C. Images of single axis tilt series were collected using a 4096 × 4096–pixel CMOS camera (TemCam-F416; TVIPS) and automated acquisition software (EM-TOOLs; TVIPS). Tilt series were acquired by a stepwise rotation of the sample from −60 to 60° in 2.0° increments. The total electron dose was limited to approximately 100 e/Å$^2$ for an individual tilt series to avoid radiation damage of the sample. Images were recorded at 300 keV, with 8.8 μm defocus, at a magnification of 30,000 × and a pixel size of 7.2 Å. An in-column Ω energy filter was used to enhance image contrast in the zero-loss mode with a slit width of 20 eV. The tilt series images were aligned and reconstructed into 3D tomograms using IMOD software (*Kremer et al., 1996*). Alignment and averaging of subtomograms were conducted by custom Ruby-Helix scripts (*Metlagel et al., 2007*) and PEET (Particle Estimation for Electron Tomography) software suite (*Nicastro et al., 2006*), using a 96 nm repeat of DMT as one particle assuming a nine-fold rotational symmetry of the axoneme. Effective resolutions were determined by Fourier shell correlation with a cutoff value of 0.143. For the visualization of tomographic slices or 3D structures, 3dmod program (IMOD software) or isosurface rendering of UCSF Chimera package (*Pettersen et al., 2004*) were used, respectively.

## Immunofluorescence microscopy

Spermatozoa in Hank's buffer were attached to the wells of an eight-well glass slide (TF0808; Matsunami). After washing out excess sperm, attached spermatozoa were briefly demembranated using 1% Nonidet P-40 for 2 min. Specimens were fixed with 2% paraformaldehyde/Hank's buffer for 10 min at room temperature, followed by treatment with cold acetone and methanol (−20°C). After rehydration with PBST (phosphate buffered saline containing 0.1% Tween-20), specimens were treated with blocking buffer (2% normal goat serum, 1% cold fish gelatin in PBST). Immunostaining was performed with anti-acetylated tubulin antibody (1:500 dilution; T6793; Sigma-Aldrich) and anti-Dnah8 antibody (1:50 dilution; generated in this study) as primary antibodies. Alexa Fluor 488 Donkey anti-mouse IgG (1:250 dilution; ab150105; Abcam) and Alexa Fluor 555 Donkey anti-rabbit IgG (1:250 dilution; ab150074; Abcam) were used as secondary antibodies with 2.5 μg/ml DAPI (Wako) for nuclear staining. Specimens were mounted with Fluoro-KEEPER Antifade Reagent (Nacalai tesque) and observed with a fluorescence microscope (BX60; Olympus) and a CCD camera (ORCA-R2; Hamamatsu).

## Kupffer's vesicle cilia analysis

Embryos developing Kupffer's vesicle were selected at 12 hpf and dechrionated before observations. To align the orientations, embryos were embedded in 0.8% of low gelling temperature agarose (Sigma-Aldrich) with 1/3 Ringer's solution. Motility of Kupffer's vesicle cilia were observed under the bright-field conditions using an inverted microscope (DMI6000B; Leica) and a high-speed camera (HAS-L1; Detect) at 1000 fps.

## RNA probe synthesis

The sequences of zebrafish PIH genes and DNAH genes were subcloned into pCRII-TOPO plasmid (Invitrogen). From the constructed plasmids, RNA probes were synthesized using SP6 or T7 RNA polymerase (Roche) with DIG RNA Labeling Mix (Roche). RNAs were purified using RNeasy Mini Kit (Qiagen). Sequences of primers used in the construction of plasmids are summarized in *Table 2*.

## Whole-mount in situ hybridization

Dechorionated embryos or dissected testes were fixed with 4% paraformaldehyde (PFA) in PBST, and then stored in methanol at −20°C. After rehydration with PBST, specimens were treated with proteinase K and re-fixed with 4% PFA/PBST solution. Hybridization was performed overnight at 63°C in hybridization buffer (750 mM NaCl, 75 mM trisodium citrate, 500 μg/ml torula tRNA, 50 μg/ml Heparin, 50% formamide, and 0.1% Tween-20) with digoxigenin-labeled RNA probes. Hybridized

**Table 2.** Primer sequences used in this study

| Purpose | Name | Sequence |
|---|---|---|
| Probes for in situ hybridization | pih1d1.F | GAACCGGAACAGTCAGTGAA |
| | pih1d1.R | TAGGGCACAAAAACGGAAAC |
| | pih1d2.F | GTGAGTCCAAGTAGGCTATT |
| | pih1d2.R | GGATAAACAAGAGTGAACTATTT |
| | ktu.F | GACGCGCTGTTCACCGGAGC |
| | ktu.R | GGATCCGCCAGCTGTGGCC |
| | twister.F | AGAGCAGGACATTCACTTCA |
| | twister.R | TTGTCTATCTTTCTCTCTGTTTCC |
| | dnah9.F | ATGTATGGAGTCTGTGGATGCAAAG |
| | dnah9.R | ACATGGCCGTACAGCAGGAA |
| | dnah9l.F | CGCGTGGACTTCATTAGAG |
| | dnah9l.R | TTTGACCGTAAGGCCTG |
| | dnah11.F | GTGTCAATATTTTTTAACAAGGTGT |
| | dnah11.R | AATATTGGACAGTTGTCGATCA |
| | dnah5.F | AAATACCTCTGTCGAATCACTG |
| | dnah5.R | ATCCTGTGGAGCCAATCC |
| | dnah8.F | ACAGAAAACGAAGAGTCTCCTC |
| | dnah8.R | ATCCATAAGGCTGCAGAAG |
| | dnah2.F | ATGGCTGACCCAGAGACC |
| | dnah2.R | CTCAGAATGGAACAGATTGTACAC |
| | dnah3.F | AGGTATTTATACTATGTCACGGAGG |
| | dnah3.R | AGTGACTTCACTTGTGTGTTTCA |
| | dnah7l.F | ATGTATTCTTTTCAGAAAGGACATC |
| | dnah7l.R | AGCATCTTCTTCAAGAATTTCCT |
| Antigen generation | pih1d1_Ag.F | ATGTCTGAGGCACTCATCATG |
| | pih1d1_Ag.R | TCACAAAGACACGACTGGC |
| | pih1d2_Ag.F | ATGGCAATCCATGGCTAT |
| | pih1d2_Ag.R | TCATAATACATTTACCGTCACATT |
| | ktu_Ag.F | ATGGACGCGGACAGACTG |
| | ktu_Ag.R | TGCTTCCAGTTGGACTGA |
| | twister_Ag.F | ATGGAGGGTCTCGCGTCC |
| | twister_Ag.R | TCAGGTCAGATTAATGCAGTC |
| | dnah8_Ag.F | TATTTCATAGGTTGTGTGTACAAG |
| | dnah8_Ag.R | TTCACGTGGCGGCAAACC |
| | dnah2_Ag.F | AGCGCCTTTATCAATGACTG |
| | dnah2_Ag.R | GATGATCTTCTTAAACTGCTCA |

DOI: https://doi.org/10.7554/eLife.36979.032

specimens were washed with 50% formamide/2 × SSCT (saline sodium citrate containing 0.1% Tween-20) followed by 2 × SSCT and 0.2 × SSCT, then treated with AP-conjugated anti-digoxigenin Fab fragments (1:4000 dilution; Roche) in blocking solution (150 mM NaCl, 100 mM maleic acid at pH 7.5, 5% blocking reagent (Roche), 5% normal goat serum, and 0.1% Tween-20) at 4°C overnight. After washing with MABT (150 mM NaCl, 100 mM maleic acid at pH 7.5, and 0.1% Tween-20), signals were developed using BM-purple (Roche). When desired intensities of staining were obtained, reactions were stopped by stopping solution (PBST containing 1 mM EDTA) and 4% PFA/PBST. Before observations, specimens were transferred into 80% glycerol/PBS to make them transparent.

For some embryos, to show stained organs clearly, posterior regions were flat-mounted or yolk was removed. Images were taken by a stereoscopic microscope (MVX10; Olympus) and a CCD camera (DP73; Olympus).

## Antibodies

Sequences encoding full length of zebrafish PIH proteins (Pih1d1, Pih1d2, Ktu, and Twister) were subcloned into the pColdI plasmid vector (Takara). Sequences encoding zebrafish Dnah8 (amino acid 895–1402) and Dnah2 (amino acid 802–1378) were subcloned into the pGEX-6P-2 plasmid vector (GE Healthcare). Recombinant polypeptides were purified from transformed *E.coli* lysate using Ni-NTA Agarose (Qiagen) for PIH proteins or Glutathione Sepharose 4B (GE Healthcare) for Dnah8 and Dnah2. Polyclonal antibodies against each purified polypeptide were raised by immunization of rabbits. Antibodies were affinity purified from serum by the antigens before use. Sequences of the primers used in the antigen production are summarized in *Table 2*. Other antibodies used are as follows: anti-Dnai1 antibody (GTX109719; GeneTex), anti-Dnali1 antibody (anti-p28 antibody; *LeDizet and Piperno, 1995*), anti-α-tubulin antibody (T9026; Sigma-Aldrich), and anti-acetylated tubulin antibody (T6793; Sigma-Aldrich).

## Immunoblot analysis

Proteins were separated by SDS-PAGE in 5–20% gradient polyacrylamide gels (Nacalai Tesque) and transferred onto polyvinylidene difluoride (PVDF) membranes (Millipore). After blocking with 5% skim milk (Nacalai Tesque) in TBST (Tris-buffered saline containing 0.1% Tween-20), membranes were incubated with primary antibodies, followed by several washes and incubation with secondary antibodies (goat anti rabbit/mouse IgG antibody peroxidase conjugated; Sigma-Aldrich). Protein signals were visualized by ECL Select Western Blotting Detection Reagent (GE Healthcare) and observed using luminescent image analyzer (ImageQuant LAS4000mini; GE Healthcare).

## Statistics

Data with biological/technical replicates are shown with mean (bar graphs)±SD (error bars). Statistical significances between WT and each mutant were tested by a two-tailed Dunnett's test, and p value < 0.05 was considered to indicate a significant difference.

## Accession numbers

The averaged subtomograms of zebrafish DMTs in this study are available at the EMDataBank (http://www.emdatabank.org/) under the following accession numbers: WT, EMD-6954; *pih1d1*$^{-/-}$, EMD-6955; *pih1d2*$^{-/-}$, EMD-6956; *ktu*$^{-/-}$, EMD-6957; *twister*$^{-/-}$, EMD-6958; *pih1d2*$^{-/-}$;*ktu*$^{-/-}$ (+OAD class), EMD-6959; and *pih1d2*$^{-/-}$;*ktu*$^{-/-}$ (-OAD class), EMD-6960.

# Acknowledgements

We would like to thank Dr T Tsukahara for the valuable discussions and experimental suggestions, Dr CH Kim for the help of zebrafish TALEN genome-editings, and Dr H Yanagisawa for the help of image processing of cryo-ET analysis. This work was supported by JST CREST Grant Number JPMJCR14M1 to M Kikkawa and JSPS KAKENHI Grant Number 16H02502 to M Kikkawa.

# Additional information

## Funding

| Funder | Grant reference number | Author |
| --- | --- | --- |
| Core Research for Evolutional Science and Technology | JPMJCR14M1 | Masahide Kikkawa |
| Japan Society for the Promotion of Science | 16H02502 | Masahide Kikkawa |

The funders had no role in study design, data collection and interpretation, or the decision to submit the work for publication.

## Author contributions
Hiroshi Yamaguchi, Conceptualization, Investigation, Visualization, Methodology, Writing—original draft; Toshiyuki Oda, Investigation, Methodology, Writing—review and editing; Masahide Kikkawa, Conceptualization, Supervision, Funding acquisition, Methodology, Writing—original draft, Project administration, Writing—review and editing; Hiroyuki Takeda, Conceptualization, Supervision, Funding acquisition, Project administration, Writing—review and editing

## Author ORCIDs
Hiroshi Yamaguchi (iD) http://orcid.org/0000-0002-8722-129X
Toshiyuki Oda (iD) http://orcid.org/0000-0001-8090-2159
Masahide Kikkawa (iD) http://orcid.org/0000-0001-7656-8194
Hiroyuki Takeda (iD) http://orcid.org/0000-0002-7932-6358

## Decision letter and Author response
Decision letter https://doi.org/10.7554/eLife.36979.050
Author response https://doi.org/10.7554/eLife.36979.051

# Additional files

## Supplementary files
• Transparent reporting form
DOI: https://doi.org/10.7554/eLife.36979.033

## Data availability
All data generated or analysed during this study are included in the manuscript and supporting files. All the electron density maps derived from cryo-electron tomography are deposited in EMD under the following accession numbers: WT, EMD-6954; pih1d1-/-, EMD-6955; pih1d2-/-, EMD-6956; ktu-/-, EMD-6957; twister-/-, EMD-6958; pih1d2-/-;ktu-/- (+OAD class), EMD-6959; and pih1d2-/-;ktu-/- (-OAD class), EMD-6960.

The following datasets were generated:

| Author(s) | Year | Dataset title | Dataset URL | Database, license, and accessibility information |
|---|---|---|---|---|
| Yamaguchi H, Oda T, Kikkawa M, Takeda H | 2018 | Doublet microtubule of zebrafish sperm axoneme, WT | http://www.ebi.ac.uk/pdbe/entry/emdb/EMD-6954/ | Publicly available at the Electron Microscopy Data Bank (accession no. EMD-6954) |
| Yamaguchi H, Oda T, Kikkawa M, Takeda H | 2018 | Doublet microtubule of zebrafish sperm axoneme, pih1d1_null mutant | http://www.ebi.ac.uk/pdbe/entry/emdb/EMD-6955/ | Publicly available at the Electron Microscopy Data Bank (accession no. EMD-6955) |
| Yamaguchi H, Oda T, Kikkawa M, Takeda H | 2018 | Doublet microtubule of zebrafish sperm axoneme, pih1d2_null mutant | http://www.ebi.ac.uk/pdbe/entry/emdb/EMD-6956/ | Publicly available at the Electron Microscopy Data Bank (accession no. EMD-6956) |
| Yamaguchi H, Oda T, Kikkawa M, Takeda H | 2018 | Doublet microtubule of zebrafish sperm axoneme, ktu_null mutant | http://www.ebi.ac.uk/pdbe/entry/emdb/EMD-6957/ | Publicly available at the Electron Microscopy Data Bank (accession no. EMD-6957) |
| Yamaguchi H, Oda T, Kikkawa M, Takeda H | 2018 | Doublet microtubule of zebrafish sperm axoneme, twister_null mutant | http://www.ebi.ac.uk/pdbe/entry/emdb/EMD-6958/ | Publicly available at the Electron Microscopy Data Bank (accession no. EMD-6958) |

| | | | | |
|---|---|---|---|---|
| Yamaguchi H, Oda T, Kikkawa M, Takeda H | 2018 | Doublet microtubule of zebrafish sperm axoneme, pih1d2, ktu, double_mutant, +OAD | http://www.ebi.ac.uk/pdbe/entry/emdb/EMD-6959/ | Publicly available at the Electron Microscopy Data Bank (accession no. EMD-6959) |
| Yamaguchi H, Oda T, Kikkawa M, Takeda H | 2018 | Doublet microtubule of zebrafish sperm axoneme, pih1d2, ktu, double_mutant, -OAD | http://www.ebi.ac.uk/pdbe/entry/emdb/EMD-6960/ | Publicly available at the Electron Microscopy Data Bank (accession no. EMD-6960) |

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
