## [Decision Letter]

Thank you for submitting your article "Systematic studies of all PIH proteins in zebrafish reveal their distinct roles in axonemal dynein assembly" for consideration by *eLife*. Your article has been reviewed by three peer reviewers, and the evaluation has been overseen by a Reviewing Editor and Anna Akhmanova as the Senior Editor. The following individual involved in review of your submission has agreed to reveal his identity: David Mitchell (Reviewer #3).

The reviewers were extremely excited by your manuscript (as was I). It is an impressive study that identifies the roles of different PIH domains in assembling different axonemal dyneins. The cryo-ET work is particularly elegant.

The reviewers had a few suggestions for improving the manuscript, which I include below.

*Reviewer #1:*

Yamaguchi et al. predicted that four genes, *pih1d1, pih1d2, ktu* and *twister*, are essential for assembly of axonemal dyneins in cilia and characterized these four gene products functionally and structurally, using their system of zebrafish and gene editing technique. They achieved two major breakthroughs in this work. One is identification of these four genes, which are essential for the assembly of outer arm dyneins and five single-headed inner arm dyneins. This finding sheds light onto the mechanism of cilia generation. The other is establishment of genetic engineering of ciliary proteins in zebrafish and its application to structural research using cryo-EM. In motile cilia research, model organisms which can be used for motile cilia research and capable for straightforward genetic engineering have been explored. The problem was that *Chlamydomonas*, the most popular species used for motile cilia research, due to abundance of motility mutants, has severe difficulty of genetic engineering, such as homologous recombination or gene editing, while *Tetrahymena*, another popular ciliated organism, has a problem of multiple copies of chromosomes. This manuscript presents probably the first work, in which mutants are designed for characterization of targeted proteins in vertebrate cilia. Both biological and technical breakthroughs deserve publication in *eLife*. This reviewer strongly recommends publication and has only a few proposals to improve this manuscript.

1) The authors claim that a fraction of cilia from *pih1d2* and *ktu* deletion mutants lack outer dyneins and these mutants lack also inner dyneins b, c and e (Figure 7A). However, in Figure 3C, there is a faint density at the loci if these inner dyneins. This reviewer guesses that these "ODA free" *pih1d2/ktu* deletion mutants still partially retain dyneins b, c and e. However, the possibility of contamination of cilia with ODA, which generate weaker density than other dyneins as seen in Figure 3C, cannot be excluded. Unless there is another basis to claim dyneins b, c and e are completely missing in these mutants, the manuscript would get benefit from more careful interpretation.

2) Regarding the above point, does one doublet, which lacks ODA (known as doublet 1 in Hoops and Witman, 1983) in *pih1d2^-/-^* and *ktu^-/-^* cilia with ODA, also lack inner dyneins b, c and e?

*Reviewer #2:*

In this manuscript, Yamaguchi et al. provide a comprehensive analysis of the potential role of different PIH1 isoforms in axonemal dynein assembly and cilia biogenesis. PIH1 is a novel Hsp90 cofactor that interacts with other proteins to form the R2TP complex. The authors identify four PIH1 isoforms in zebrafish, namely: Pih1d1, Pih1d2, Ktu, and Twister. They carry out a comprehensive series of experiments to show the localization of these proteins, their effect on sperm motility, their effect on axonemal dynein assembly by studying the doublet microtubule structure by cryo-ET, and their effect on Kupffer's vesicle cilia.

The paper is well written, and the experiments are very comprehensive. The most important conclusion from this study is that indeed al Pih1 isoforms play some roles in axonemal dynein assembly. However, it is more of a phenotypic study and does not provide insights into the exact assembly roles of these different Pih1 isoforms. What does Pih1 interact with to promote this assembly? Where is the assembly occurring? How are the different isoforms distinguished in their activities? Etc. Minor comment: The antibodies generated for the different Pih1 isoforms are polyclonal. I am surprised that there is no cross reactivity of the different antibodies (Figure 1C).

*Reviewer #3:*

Yamaguchi et al. have taken a comprehensive look at the role of PIH-domain proteins in zebrafish. Although several PIH proteins in various model systems have been linked to assembly of axonemal dyneins in the cytoplasm, prior to their transport into cilia, some PIH proteins have also been implicated in the assembly of other macromolecular complexes including pre-rRNA splicing SNORPs and telomerase. The divergence of PIH protein sequences has made it difficult to draw conclusions about the conservation of function among paralogs within a given organisms, or apparent homologs between organisms. This paper does a wonderful job of beginning to sort out the functions of these paralogs.

In situ hybridization was used to show the distribution of expression of each PIHN protein in fish embryos at two developmental stages, providing clear patterns with interesting relationships to ciliated tissues. Importantly, isoforms specific antibodies were generated that confirmed that all four are expressed in sperm and are cytoplasmic, rather than axonemal. These antibodies also confirmed that expression of each protein was disrupted by Talen and CRISPR generated mutants.

Motility of sperm analyzed from high speed video revealed only minor effects of the loss of PIH1D1, PIH1D2 or Ktu, but dramatic effects of loss of Twi. Loss of PIH1D1 also had little effect on KV cilia, whereas PIH1D2 and Ktu had greater effects and Twi loss was the most dramatic.

Effects on motility were then characterized at the level of axonemal ultrastructure and by immunofluorescent and blot analysis with antibodies specific to four zebrafish dynein subunits. This careful work provided clear links between the abnormal assembly patterns of specific dyneins and the resulting motility phenotypes. Apparent tissue-specific effects of PIH proteins was correlated with tissue-specific expression patterns of dynein genes. The analysis of axonemal dyneins is especially difficult in metazoa due to the long evolutionary distance from the most well-studied system (*Chlamydomonas*), and this manuscript provided an excellent start to understanding the diversity of dynein isoforms in different fish tissues.

The added analysis of a double mutant between PIH1D2 and Ktu, while useful to show that each contributes differently to dynein assembly as well as being potentially redundant for some assembly processes, did not unfortunately contribute mechanistic insight, but did reveal a differential effect on outer dynein arms in the proximal and distal half of sperm axonemes.

One recommendation: when describing the analysis of the many confusing inner dynein arm isoforms, it would be useful to make reference to the very careful and comprehensive analysis of dynein phylogeny and evolution by Martin Kollmar (2016). While it may be useful to number the single-headed IDAs the same way that they are numbered in *Chlamydomonas*, the actual molecular correspondence between dynein isoforms in vertebrates and *Chlamydomonas* is more complex, including the swapping of N-terminal tail sequences on some of these dyneins in all the viridaplantae.

---

## [Author Response]

Reviewer #1:

[…] This reviewer strongly recommends publication and has only a few proposals to improve this manuscript.1) The authors claim that a fraction of cilia from pih1d2 and ktu deletion mutants lack outer dyneins and these mutants lack also inner dyneins b, c and e (Figure 7A). However, in Figure 3C, there is a faint density at the loci if these inner dyneins. This reviewer guesses that these "ODA free" pih1d2/ktu deletion mutants still partially retain dyneins b, c and e. However, the possibility of contamination of cilia with ODA, which generate weaker density than other dyneins as seen in Figure 3C, cannot be excluded. Unless there is another basis to claim dyneins b, c and e are completely missing in these mutants, the manuscript would get benefit from more careful interpretation.

The comment is reasonable and we cannot exclude the partial retaining of IAD b, c, and e in *pih1d2*^-/-^;*ktu*^-/-^ -OAD DMTs, although our isosurface rendering image showed missing of theses IADs because of the different threshold. We revised the interpretation of our data as follows:

“The +OAD class possessed a full set of axonemal dyneins, except for a smaller IAD c, like the *ktu*^-/-^ axoneme (Figure 3J). […] However, note that the -OAD class showed faint densities of these IADs in the subtomographic slice (Figure 3—figure supplement 1C), which suggests that IAD b, c, and e were retained partially in the -OAD class axonemes.”

We also modified Figure 7A to be consistent with the revised interpretation.

2) Regarding the above point, does one doublet, which lacks ODA (known as doublet 1 in Hoops and Witman, 1983) in pih1d2^-/-^ and ktu^-/-^ cilia with ODA, also lack inner dyneins b, c and e?

In zebrafish sperm, Zhang et al. (2014) reported that all DMTs retained OADs, and our cryo-ET analysis and conventional TEM observation also support this claim (Author response image 1). Although *Chlamydomonas* flagella contain one DMT which lacks OADs (Hoops and Witman, 1983; Bui et al., 2012), this feature is not always the case in other organisms: in *Tetrahymena* cilia, all the DMTs retain OADs (Pigino et al., 2012); in sea urchin sperm flagella, one DMT retain unique OAD-like linker structures (Lin et al., 2012).

**Author response image 1. respfig1:** TEM image of zebrafish sperm axoneme.

To describe this point, we revised the manuscript as follows:

“The axoneme of zebrafish sperm had the characteristic 9+2 arrangement of DMTs surrounding central-pair microtubules (Figure 3—figure supplement 1A). To analyze the structure of DMTs in more detail, subtomographic averaging was applied using the 96-nm repeat of DMTs assuming nine-fold rotational symmetry of the axoneme, since we did not detect any obvious heterogeneity of nine DMTs in zebrafish sperm unlike *Chlamydomonas* flagella and sea urchin sperm (Hoops and Witman, 1983; Bui et al., 2012; Lin et al., 2012).”

Reviewer #2:

[…] What does Pih1 interact with to promote this assembly? Where is the assembly occurring? How are the different isoforms distinguished in their activities? Etc. Minor comment: The antibodies generated for the different Pih1 isoforms are polyclonal. I am surprised that there is no cross reactivity of the different antibodies (Figure 1C).

Our polyclonal antibodies against PIH proteins were raised by immunization of rabbits with full length of recombinant zebrafish PIH proteins as antigens. We assessed the specificities of these antibodies (Figure 1–figure supplement 1E) and they actually showed no detectable cross reactivity. We think two points of our method below may contribute to the lower background of immunoblot:

1) Before use, antibodies were affinity purified from the rabbit serum using antigens-attached PVDF membranes.

2) When visualizing signals, we used ECL Select Western Blotting Detection Reagent. Since this reagent produces very high signal intensity, we could reduce the amount of sample loading and primary antibodies.

These points are described in the “Materials and Methods” section of the manuscript.

Reviewer #3:

[…] One recommendation: when describing the analysis of the many confusing inner dynein arm isoforms, it would be useful to make reference to the very careful and comprehensive analysis of dynein phylogeny and evolution by Martin Kollmar (2016). While it may be useful to number the single-headed IDAs the same way that they are numbered in Chlamydomonas, the actual molecular correspondence between dynein isoforms in vertebrates and Chlamydomonas is more complex, including the swapping of N-terminal tail sequences on some of these dyneins in all the viridaplantae.

Thank you for your careful reading of our manuscript. The recommendation is helpful. For a better understanding of dynein isoforms in zebrafish, we made a table that shows the gene correspondence of dynein heavy chain isoforms among zebrafish, human, and *Chlamydomonas* (Table 1).

We also revised the manuscript as follows:

“Zebrafish have three OAD β-HC genes: *dnah9, dnah9l*, and *dnah11*, and two OAD γ-HC genes: *dnah5* and *dnah8*. […] The gene correspondence of dynein heavy chains among zebrafish, human, and *Chlamydomonas* are summarized in Table 1, based on the comprehensive analysis of dynein phylogeny by Kollmar (2016).”